**Synoptic fluctuation of the Taiwan Warm Current in winter on the East China Sea shelf**
Jiliang Xuan[1], Daji Huang[1, 2], Thomas Pohlmann[3], Jian Su[3], Bernhard Mayer[3], Ruibin Ding[2,1], Feng Zhou[1, 2]
[1]State Key Laboratory of Satellite Ocean Environment Dynamics, Second Institute of Oceanography,
State Oceanic Administration, Hangzhou, China
[2]Ocean College, Zhejiang University, Zhoushan, China
[3]Institute of Oceanography, University of Hamburg, Hamburg, Germany
*Correspondence to:* Daji Huang (djhuang@sio.org.cn)
**Highlights**
● Synoptic fluctuations of the wintertime Taiwan Warm Current appear mainly in two areas: north of
Taiwan and the inshore area
● Synoptic fluctuation is mainly driven by the Taiwan Strait Current north of Taiwan and by wind in
the inshore area
● Large Taiwan Strait Current intrusion generates a cross-shore transport from the coastal area to the
offshore area
● Winter monsoon affects the alongshore transport of Taiwan Warm Current water between the 30
and 100 m isobaths
● Winter monsoon affects the cross-shore transport of Taiwan Warm Current water at the latitudes
26.5 N and 28 N
**Abstract.** The seasonal mean and synoptic fluctuation of the wintertime Taiwan Warm Current (TWC)
were investigated using a well validated finite volume community ocean model. The spatial distribution
and dynamics of the synoptic fluctuation were highlighted. The seasonal mean of the wintertime TWC
has two branches: an inshore branch between the 30 and 100 m isobaths and an offshore branch between
the 100 and 200 m isobaths. The Coriolis term is much larger than the inertia term and is almost balanced
by the pressure gradient term in both branches, indicating geostrophic balance of the mean current. Two
areas with significant fluctuations of the TWC were identified during wintertime. One of the areas is
located to the north of Taiwan with velocities varying in the cross-shore direction. These significant
cross-shore fluctuations are driven by barotropic pressure gradients associated with the intrusion of the
Taiwan Strait Current (TSC). When a strong TSC intrudes to north of Taiwan, the isobaric slope tilts
downward from south to north, leading to a cross-shore current from the coastal area to the offshore area.
When the TSC intrusion is weak, the cross-shore current to the north of Taiwan is directed from offshore
to inshore. The other area of significant fluctuation is located in the inshore area between the 30 and 100
m isobaths. The fluctuations are generally strong both in the alongshore and cross-shore directions, in
particular at the latitudes 26.5 ˚N and 28 ˚N. Wind affects the synoptic fluctuation through episodic events.
When the northeasterly monsoon prevails, the southwestward Zhe-Min Coastal Current dominates the
inshore area associated with a deepening of the mixed layer. When the winter monsoon is weakened or
the southwesterly wind prevails, the northeastward TWC dominates in the inshore area.
**Keywords:**
Synoptic fluctuation, East China Sea, Taiwan Warm Current, Taiwan Strait Current, Kuroshio

## 1 Introduction

On the East China Sea (ECS) shelf, the mean path of the Taiwan Warm Current (TWC) has two branches: the inshore branch along the 50 m isobath and the offshore branch along the 100 m isobath (Su and Pan, 1987). The summer TWC has been well studied because the current is stationary and strong, with an average speed of 0.3 m/s (Guan, 1978; Fang et al., 1991; Isobe, 2008; Yang et al., 2011, 2012). The spatial structure and temporal variation of the wintertime (December to March) TWC are less known due to its weak mean surface velocity, according to a climatological structure of the surface current in the ECS mapped by Qiu and Imasato (1990).

The wintertime TWC on the ECS shelf shows synoptic fluctuations (Cui et al., 2004; Zhu et al., 2004; Zeng et al., 2012; Huang et al., 2016). These synoptic fluctuations show some features common with those over other continental shelves, i.e., they have periods between 3 and 15 days and are associated with coastal sea level changes, which can be explained by local winds or by coastal trapped waves (Huyer, 1990; Brink, 1991; Huthnance et al., 1986). Huang et al. (2016) have shown that the wind was a main physical factor which caused the temporal variation of the wintertime currents at the synoptic scale in the coastal area of the ECS. However, the dominant physical factors of the TWC fluctuations still lack study; the fluctuations on the whole shelf of the ECS may be complicated due to the complex bottom topography, alternating wind forcing and conjunction of several current systems such as the Kuroshio Current, the Taiwan Strait Current (TSC) and the Zhe-Min Coastal Current (ZMCC). These synoptic fluctuations are also known to influence the regional material transport, especially when the amplitude of the fluctuations is comparable to, or even larger than, the mean current. On the ECS shelf, some recent

observations have shown that the TWC has an episodic wintertime feature (Zhu et al., 2004) and the
variations of the TWC in winter have an amplitude as large as 0.2 m/s (Zeng et al., 2012). Moreover, it
has been observed that the variations of the TWC in winter cause a cross-shore current which is closely
linked to the alongshore component (Huang et al., 2016). Therefore, we focus on studying the spatial
patterns of synoptic fluctuations to better understand the role of the wintertime TWC on the cross-shore
water exchange.

A comparison between the wintertime climatological density (Fig. 1a) and synoptic density distributions
observed during two surveys (Figs. 1b and 1c) suggests that two distinct areas with significant synoptic
fluctuations exist. The climatological density is taken from the Generalized Digital Environment Model
(GDEM, Carnes, 2009) data, and the two surveys were carried out in February 2007 by two research
vessels. Because the isopycnal lines are closely related to geostrophic currents, we can infer the strength
of the TWC from the horizontal gradient of the isopycnals between $24$-$\sigma_t$ and $25$-$\sigma_t$ contours (Fig. 1a).
This accounts for the fact that in winter the water mass of TWC is located in this density range [according
to the hydrography analysis of Su et al. (1994)]. The two-branch structure of the TWC can be inferred
from the wintertime climatological density. In this paper, we defined that the near-coast area is the area
between the coast and 30 m isobath where the ZMCC occurs; the inshore area is the area between the 30
and 100 m isobaths where the TWC inshore branch dominates; and the offshore area is the region between
the 100 and 200 m isobaths where the TWC offshore branch prevails. According to the hydrographic
data analysis and numerical interpretation by Su and Pan (1987), the TWC inshore and offshore branches
mainly occur close to those specific isobaths. However, these two branches were missing during the two
synoptic surveys (Figs. 1b and 1c), indicating strong synoptic fluctuations of the TWC on the ECS shelf.
Furthermore, the density anomalies between the two surveys and the GDEM data (Figs. 1d and 1e)
indicate that the most significant fluctuations are located north of Taiwan and in the inshore area. Both
surveys show negative density anomalies north of Taiwan, indicating that the TWC was weak and that
more low-density coastal water was transported to the ECS shelf during the observational periods. The
density anomalies in the inshore area show different patterns for the two synoptic surveys, with a positive
anomaly in the first survey (Fig. 1d) and a negative anomaly in the second (Fig. 1e), indicating a strong
synoptic fluctuation in the inshore area.

Figure 1

Candidate factors for driving these synoptic fluctuations are local wind, surface cooling, and the upstream
currents of the Kuroshio Current and the TSC. As discussed by Huyer (1990), wind is often considered
as the major driving mechanism of synoptic fluctuations of the wintertime TWC. The northeasterly
monsoon wind in winter blows against the northeastward TWC and produces a southwestward ZMCC
(Chuang and Liang, 1994; Oey et al., 2010). Zhu et al. (2004) suggested that the occurrence and duration
of the TWC are associated with the meandering of the Kuroshio Current north of Taiwan. The
northeastward TSC, as an upstream flow of the TWC, also influences the synoptic fluctuation of the
wintertime TWC. Hong et al. (2011) and Hu et al. (2010) summarized that the temporal and spatial
variation of TSC is modulated by strong wind forcing, complex topography and circulation in the
northern South China Sea as well as coastal water input and the Kuroshio intrusion. Guan and Fang (2006)
showed evidence that the TSC and the TWC merge in the area between the Taiwan Strait and the Zhe-
Min coastal region. Takahashi and Morimoto (2013) pointed out that the temporal variation of the TWC
is characterized by the propagation of vorticity anomalies originating from northeast of the Taiwan Strait,
which further demonstrated that the fluctuations of TWC was associated with its upstream currents such
as the TSC.

To explore the spatial distribution of synoptic fluctuations of the wintertime TWC on the ECS shelf,
current data with high resolution in both space and time are required. Previous studies on the wintertime
TWC were based on cruise surveys (Su and Pan, 1987; Chen et al., 1994; Chen and Wang, 1999),
anchored mooring observations (Zhu et al., 2004; Zeng et al., 2012; Huang et al., 2016) and numerical
simulations (Guo et al., 2003, 2006; Yang et al., 2011, 2012; Xuan et al., 2012, 2016). The observation
data are limited in terms of temporal and spatial coverage; hence, they cannot fully reveal the synoptic
fluctuations of the TWC and their regional differences. Numerical simulations provide a promising
approach for studying the overall structure and driving mechanisms of synoptic fluctuations of the TWC
in more detail.

In this study, the Finite Volume Coastal Ocean Model (FVCOM; Chen et al., 2003) is used to investigate
wintertime TWC synoptic fluctuations and their mechanisms. The rest of this paper is organized as
follows. In Sect. 2, we provide a description of methods and validation. The mean distribution, synoptic
fluctuations, and dynamic diagnostics of the wintertime TWC are given in Sect. 3. The impact of synoptic
fluctuation on water exchange is further discussed in Sect. 4, followed by conclusions in Sect. 5.

**2 Methods and validation**
**2.1 Model configuration**
To investigate the currents (TWC, Kuroshio Current, ZMCC, etc.) and their synoptic fluctuations on the
ECS shelf, a 3-D unstructured-grid (Fig. 2, left panel) FVCOM is developed for the entire Bohai, Yellow,
and East China Seas (part of the Japan/East Sea, and part of the Pacific Ocean). A regional refinement of
the resolution (approximately 3 km) is specified around the ECS shelf break at the 200 m isobaths, where
a strong excursion of the Kuroshio Current also occurs. The General Bathymetric Chart of the Oceans
(GEBCO) provides high-resolution (approximately 1 km) bathymetric data (Smith and Sandwell, 1997).
Twenty vertical layers with 76954 triangle cells were specified in the water column in a sigma-stretched
coordinate system.

The driving forces of the numerical simulation include tides, river discharge, surface heat fluxes, wind,
and open boundary conditions. Harmonic constants of 11 major tidal constituents ($M_2$, $S_2$, $N_2$, $K_2$, $K_1$, $O_1$,
$P_1$, $Q_1$, $M_4$, $MS_4$, and $MN_4$) were used; these are based on the Oregon State University global inverse
tidal model TPXO.7.0 (Egbert et al., 1994; Egbert and Erofeeva, 2002). The daily-mean river discharge
of the Changjiang and Huanghe were taken from publicly available observation data at the Datong
hydrometric station (http://yu-zhu.vicp.net/). Other rivers were not included because of their small
discharges, e.g., the Qiantang River, with the largest runoff from the Zhejiang coast, has a climatological
mean discharge in winter of about 230 $m^3$/s, which is nearly negligible compared to the Changjiang
winter discharge of about 11500 $m^3$/s. The daily-mean heat fluxes were from the objectively analyzed
air–sea fluxes (Yu and Weller, 2007), and the 3-hourly wind stress and 10 m wind speed data was from
the ERA-40 re-analysis (Uppala et al., 2005). The open boundary conditions, including daily temperature,
salinity, and fluxes at the Taiwan Strait, the western Pacific Ocean, and the Japan/East Sea, were obtained
from the Hybrid Coordinate Ocean Model (Bleck, 2002) and interpolated onto the FVCOM model grid
points. The temporal resolution of all the driving force fields is better than or equal to one day, which is
essential to resolve synoptic fluctuations.

The hindcast outputs of sea surface height, temperature, salinity, and velocities for the five years of
simulation from 2009 to 2013 are used, following three spin-up years (2006-2008) initiated with the
temperature and salinity taken from the Hybrid Coordinate Ocean Model and velocity set to zero. The
initial conditions are ramped-up over a period of 30 days and at the lateral boundaries a sponge layer was
used with the same method as Chen et al. (2008). The model time step was 15 seconds for the 2-D
barotropic mode and 90 seconds for the 3-D baroclinic mode. All of the output fields were processed
with a tidal filter (Godin, 1972) to remove tidal oscillations (considering that the major time scale of
synoptic fluctuations in this study area is 3–15 days).

Since the currents in 2009 could partly be validated by means of available observational data (see Sect.
2.2), the currents from January 1 to February 28, 2009 were selected for analysis of the wintertime TWC.

**2.2 Validation of the mean currents and synoptic fluctuations**
The mean currents, e.g., the Kuroshio Current, the TWC, and the ZMCC, were calculated by averaging
the outputs of January and February 2009. We validated the mean currents in terms of circulation
structure, boundary fluxes, and coastal currents.

The FVCOM has reproduced almost all of the known circulation structure in the ECS in winter. The
surface mean currents (Fig. 2) shows three major currents: the Kuroshio Current, the TWC, and the
ZMCC. The Kuroshio Current, with a speed of about 1 m/s, enters the ECS just northeast of Taiwan and
flows along the shelf break up to the northern area and ultimately leaves the ECS through the Tokara
Strait. Both the route and strength of the Kuroshio are comparable with those reported in the literature
(Guan, 1978; Qiu and Imasato, 1990). The TWC has two northeastward branches, one inshore (between
the 30 and 100 m isobaths) and another offshore (between the 100 and 200 m isobaths), which is
consistent with Su and Pan (1987). The southwestward directed ZMCC in the nearshore area from the
Changjiang Estuary to the Taiwan Strait agrees well with that reported in previous studies (Guan and
Mao, 1982; Zeng et al., 2012).

The simulated volume transports across the Taiwan Strait, the East Taiwan Channel, the Tsushima Strait,
the Tokara Strait, and the shelf break of the 200 m isobath were validated using results from the literature
(Table 1). The simulated transports were accurate enough to reproduce volume transport (1.22 Sv)
through the Taiwan Strait which is closer to the observation value (1.20 Sv) from Isobe (2008) than
former model results. The volume transports across the Taiwan Strait and the Tokara Strait, and the cross-
shore exchange, affected the path and magnitude of the TWC. The annual mean transport across the 200
m isobath toward the shelf is 1.66 Sv, which is balanced by the inflow from the Taiwan Strait (1.22 Sv)
and the outflow through the Tsushima Strait (2.85 Sv).


Figure 2


| Table 1

Figure 3 shows a comparison between simulation and observation results for the alongshore currents and
the cross-shore currents on the ECS shelf. The observational data were obtained from four mooring
surveys (Fig. 2, red stations) off the Zhe-Min coast (Zeng et al., 2012). The observed and simulated
currents were both averaged for the observational period, which was from January 1 to February 28,
2009. Using the same method as in Huang et al. (2016), we defined the positive alongshore current
direction as from southwest (218 °) to northeast (38 °), which is the mean tangential direction of the
isobaths on the southwestern shelf of the ECS. The positive cross-shore direction is from northwest (308 °)
to southeast (128 °), normal to the isbaths. The alongshore components (Figs. 3a and 3b) show that the
ZMCC flows southwestward parallel to the coast in winter, with a maximum speed of 0.15 m/s along the
30 m isobath. The TWC flows northeastward with a speed of 0.05 m/s, and the core is located in the
lower layer at about 50 m at Station 4. The cross-shore component (Figs. 3c and 3d) is much weaker than
the alongshore components, and it shows a complex spatial pattern. It flows offshore in the upper layer
and onshore in the lower layer at Station 1. Moreover, it mainly flows onshore at Station 2, and it flows
offshore in the entire water column at Stations 3 and 4. Altogether, the simulated pattern and magnitude
both of the alongshore and cross-shore components are in good agreement with the observations.
However, there are some differences between the observed and simulated results; for example, the
simulated ZMCC occupies a broader space than that in the observations. This may have been caused by
the relatively low number of observational stations.

| Figure 3

Synoptic fluctuations of the TWC inshore branch during January and February 2009 were also validated
against the mooring results (Fig. 4). Since the TWC shows a strong signature at Station 4, the time series
of the alongshore currents and cross-shore currents in the whole water column of Station 4 were used for
the validation. To eliminate the influence of local effects, the simulated currents were averaged in a $10 \times$
$10$ km$^2$ area around Station 4. Both the observed and simulated results show that the TWC fluctuates
with a period of 3–15 days. The simulated TWC (Fig. 4a, warm color) appeared stronger (> 0.1 m/s) on
Jan. 7, Jan. 12, Jan. 18, Jan. 21, Jan. 26, Jan. 29, Feb. 10, Feb. 14, Feb. 19, Feb. 22, and Feb. 25, which
agrees well with data from the observations (Fig. 4b). The time series of the simulated cross-shore
component (Fig. 4c) are virtually in phase with the observations (Fig. 4d). The magnitude of the cross-
shore fluctuations is comparable to the alongshore fluctuations. This is different to the anisotropic
characteristic of the mean currents (Fig. 3), for which the alongshore component is nearly one order of
magnitude larger than the cross-shore component.

Figure 4

**2.3 EOF analysis of synoptic fluctuations**
The Empirical Orthogonal Function (EOF) method (Emery and Thomson, 2001), as a statistical method,
has been used to understand synoptic fluctuations of the wintertime TWC. The simulated currents from
Jan. 1 to Feb. 28, 2009 were selected and their anomalies were calculated. Then, using the Matlab EOF-
function, the current vectors were separated into several orthogonal modes to show the spatial and
temporal variations. Because the first two leading modes explain 91 % of the total variance, only these
two modes were used for the analysis.

The spatial distributions of the two leading EOF modes were used to analyze the regional difference of
the synoptic fluctuations. To investigate the driving force of the two EOF modes, the temporal variation
was compared to the potential influence factors, such as wind, upstream currents, and net surface heat
flux.

**2.4 Momentum analysis**
The driving mechanisms of the synoptic fluctuations were further analyzed using the momentum
equation. First, the momentum balance as implemented in FVCOM (Chen et al., 2003) is shown in Eq.
(1). The three terms on the left hand side represent local acceleration, Coriolis acceleration, and advection,
respectively, and the three terms on the right hand side represent pressure gradient, friction, and diffusion,
respectively.
$$\frac{\partial \vec{V}}{\partial t} - 2\vec{\Omega} \times \vec{V} + (V \cdot \nabla \vec{V}) = -\frac{1}{\rho_0} \nabla P + \frac{\partial}{\partial z}(K_m \frac{\partial \vec{V}}{\partial z}) + \vec{F} , \qquad (1)$$

where $\vec{V}$ is velocity, $\vec{\Omega}$ is the Earth's rotation angular velocity, $\rho_0$ is the average density, $P$ is
pressure, $K_m$ is the vertical eddy viscosity coefficient, and $\vec{F}$ is horizontal diffusion.

Second, according to the hydrostatic approximation used in FVCOM [as shown in Eq. (2)], the pressure
gradient is given as the product of density times the gravitational acceleration. This results in Eq. (3),
which indicates that pressure gradient can be decomposed into the effects of the barotropic and baroclinic
components, as shown in Eq. (4).

$$\frac{\partial P}{\partial z} = \rho g \,, \qquad (2)$$


$$P_z = \int_z^\eta \rho g dz = \int_z^\eta (\rho_0 + \rho') g dz = \rho_0 g(z+\eta) + \int_z^\eta \rho' g dz \,, \qquad (3)$$


$$\nabla \vec{P} = \rho_0 g \nabla \eta + \nabla(\int_z^\eta \rho' g dz), \qquad (4)$$


where $\rho$ is density, $\rho'$ is density anomaly, $g$ is the gravitational acceleration, and $\eta$ is sea surface
height.

Finally, the momentum equation is vertically integrated to estimate momentum balance for the water
column. Since the horizontal diffusion is a comparatively small term, it is neglected for simplicity.
$$\underbrace{\int_{-H}^0 \frac{\partial \vec{V}}{\partial t}}_{Acceleration} + \underbrace{\int_{-H}^0 -2\vec{\Omega}\times\vec{V}}_{Coriolis} + \underbrace{\int_{-H}^0 (V\bullet\nabla\vec{V})}_{Advection} = \underbrace{\underbrace{-gH\nabla\eta}_{Barotropic} - \underbrace{\int_{-H}^0 \nabla(\int_z^\eta \rho' g dz)}_{Baroclinic}}_{Total\,Pressure} + \underbrace{\rho_a C_D \left|\vec{U}\right|\vec{U}}_{\tau_a} - \underbrace{k_b \left|\vec{V_b}\right|\vec{V_b}}_{\tau_b}, \qquad (5)$$

where $\tau_a$ is wind stress and $\tau_b$ is bottom stress, $\rho_a$ is the density of air, $\vec{U}$ is the wind speed at 10 m
above sea surface, $C_D$ is a drag coefficient at the sea surface (which varies with wind speed $\vec{U}$ ), $k_b$
is a bottom friction coefficient ( $k_b$ =0.005), and $\vec{V_b}$ is the simulated velocity at the bottom.

**3 Results**
**3.1 Mean distribution of TWC in winter**
Since the observational results (Su and Pan, 1987; Zeng et al., 2012) show that both branches of the
wintertime TWC are flowing in the subsurface, we use the vertical maximum velocity (VMV) and its
corresponding depth as two indices to quantify the strength of the subsurface currents (Fig. 5).

As stated above, the distribution of the VMV shows two branches of the TWC (Fig. 5a). The inshore
branch (Fig. 5a, blue arrow of IB), which was located between the 30 and 100 m isobaths, followed a
straight route from the northwest of Taiwan to the northern ECS shelf. The offshore branch (Fig. 5a, blue
arrow of OB) existed near the 100 m isobath and had two meanders. The two meanders turn to the cross-
shore direction along latitudes 26.5 °N and 28 °N. These two branches are further illustrated in the
distributions of current speed along the six cross-TWC sections (S1-S6), which were located at critical
points in the two meanders (Fig. 6). From the VMV structure, it can be inferred that the intrusions of the
TSC and the Kuroshio Current both affected the origin of the offshore branch (Fig. 6, S1–S3).

We further examined the subsurface current core using the depth of the VMV (Fig. 5b). We found that
the VMV of the TWC was located 40–60 m below the surface at the inshore branch and 20–40 m below
the surface at the offshore branch. Figure 6 shows the VMV positions in the subsurface layer; it also
illustrates that the depth of the subsurface VMV in the inshore branch was deeper than that in the offshore
branch. The difference can be explained by the combined effects of baroclinicity and wind friction.
Assuming a relatively spatially homogeneous heat loss, different cooling occurs, due to the smaller heat
capacity of the shallow coastal water compared to the deeper offshore waters; hence generating a
northwestward horizontal density gradient leading to a northeastward thermal current (vertical current
shear) according to the thermal wind relationship, resulting in an upward-increasing northeastward flow.
The northeasterly wind in winter weakens the northeastward TWC, particularly in the upper layer, which
leads to the formation of the subsurface VMV. Therefore, the fact that the depth of the subsurface current
core in the inshore branch is greater than that in the offshore branch indicates weaker baroclinicity or
stronger wind friction on the inshore branch than on the offshore branch.

The magnitude of the wintertime TWC was obtained by flux analysis. Two dividing lines (Fig. 5a, red
lines) were defined as the boundaries for the ZMCC, the TWC inshore branch, and the TWC offshore
branch, which had the weakest flows. The flux of each branch (Fig. 5c) was calculated using the
horizontal integration between the boundaries and the vertical integration in the water column. The
inshore branch intensifies along its way and becomes significant north of 26.5 °N, showing particularly
strong flow velocities between 27.5 and 28.0 °N. In this area, the subsurface current was much stronger
from S4 to S5 than in the other areas (Fig. 6). The flux in the entire offshore branch was large, particularly
north of Taiwan.

| Figure 5 |

| Figure 6 |

**3.2 Synoptic fluctuations**
The observations (Fig. 4) have demonstrated that the synoptic fluctuation in the TWC inshore branch
(near 121.5 °E, 27.0 °N) is significant. We further investigated the regional difference of fluctuations in
the two TWC branches in winter 2009 using the following three steps: (i) two regions with significant
fluctuations are identified by the current standard deviations of the VMV (Fig. 7) and the corresponding
temporal variation of vertical structures at their extremes (Fig. 8); (ii) each of the two significant
fluctuations is decomposed into EOF components (Fig. 9), and (iii) the influence factors, such as wind,
upstream currents, and net surface heat flux, are investigated by examining their correlations with the
first two leading EOF components (Figs. 10 and 11).

The current standard deviations (Fig. 7) shows that prominent fluctuations occurred in two regions: north
of Taiwan and the inshore area. The standard deviations of VMV at the two regions were larger than 0.1
m/s (comparable to the mean currents). In the area north of Taiwan, the fluctuation was located in the
origin area of the TWC offshore branch. The fluctuation in this region was in phase with the fluctuation
in the Taiwan Strait, indicating that the TSC played an important role in generating the fluctuation north
of Taiwan (to a greater extent than did the Kuroshio intrusion). The TWC fluctuation had a strong cross-
shore component, which means the fluctuation transported the water north of Taiwan to both the inshore
and offshore branches. In the inshore area, the fluctuations were influencing a wide region between the
30 and 100 m isobaths, with a magnitude that was sometimes larger than the mean flow (Fig. 5a). These
strong fluctuations led to an episodic occurrence of the TWC inshore branch, as observed at the site off
the Zhe-Min coast (Fig. 4, red color). When the TWC inshore branch was weakened due to these
fluctuations, the ZMCC might even dominate a wide region out to the 100 m isobath, especially at the
surface (Fig. 4, blue color).

| Figure 7 |

The vertical structures of the fluctuations north of Taiwan and in the inshore area at two representative
points and their relation with upper mixed layer depth are further analyzed (Fig. 8). The major component
(the alongshore current) of the TWC in each of the two regions (P1 and P2, Fig. 7) is used to show the
vertical structure of the fluctuation. The depths of the upper mixed layer were determined by a
Richardson number criterion (Mellor and Durbin, 1975; Grachev et al., 2013; Richardson et al., 2013),
i.e., where the critical Richardson number equals 0.25 in this paper [as in Xuan et al. (2012)]. The mean
depth of the upper mixed layer north of Taiwan (20 m) was much shallower than the mean depth in the
inshore area (42 m). However, the TWC (Fig. 8, warm color) fluctuated with significant variations of the
upper mixed layer depth (Fig. 8, gray lines) in both areas. When the upper mixed layer deepened, the
northeastward TWC (Fig. 8, warm color) was weakened or even replaced by the southwestward ZMCC,
and vice versa. Wind and surface cooling, which both drive the mixed layer depth, can affect the TWC
fluctuation.

| Figure 8 |
       |---|


The TWC fluctuations were further decomposed into EOF modes. The first two leading EOF modes
account for 54% and 37% of the total variances (Fig. 9), associated with the two prominent fluctuations
north of Taiwan and in the inshore area (Fig. 7). Both EOF modes had a maximum fluctuation larger than
0.2 m/s (comparable to the mean currents). The spatial pattern of the first EOF mode (EOF1, Fig. 9a)
shows that the fluctuation continued from the Taiwan Strait to the area north of Taiwan, indicating that
the fluctuation north of Taiwan was related to the TSC and not to the Kuroshio Current. The alongshore
component also showed a strong fluctuation in the Taiwan Strait, which means that the TSC episodically
intruded the shelf. The cross-shore component revealed a fluctuation north of Taiwan that was larger than
0.1 m/s. This cross-shore fluctuation impacted on the trajectory of the TWS water, synoptically flowing
into the TWC inshore branch, offshore branch, or Kuroshio Current.

The spatial pattern of the second EOF mode (EOF2, Fig. 9b) shows a synoptic fluctuation in the inshore

area. The area with alongshore fluctuation (Fig. 9d) larger than 0.1 m/s was located between the 30 and

100 m isobaths, which demonstrates that the TWC could episodically affect this area. In addition, there

were cross-shore fluctuations in the inshore area (Fig. 9f), mostly along the latitudes 26.5 N and 28 N.

The latitudes of larger cross-shore fluctuations agreed well with the latitudes where the TWC offshore

branch of the mean currents (Fig. 5a) turned to the cross-shore direction. This indicated that the cross-

shore transports were most significant at the latitudes 26.5 N and 28 N, according to both the mean

currents and the synoptic fluctuations.

Figure 10 shows the temporal variation of EOF1 and its relation with north-south component of wind

speed, net surface heat flux, the TSC, and the Kuroshio Current. We found a close correlation between

EOF1 and TSC (R = 0.86), demonstrating that the TSC played the most important role in generating the

TWC fluctuation north of Taiwan. The EOF1 and TSC were positively correlated, meaning that a larger

TSC intrusion north of Taiwan leads to a cross-shore current from the coastal area to the offshore area

and that a weak TSC intrusion causes a cross-shore current from offshore to inshore north of Taiwan.

Figure 11 shows the temporal variation of EOF2 and its relation with the north-south component of wind

speed, net surface heat flux, the TSC, and the Kuroshio Current. It can be seen that EOF2 and wind are

well correlated (R = 0.89), indicating the important role of wind in generating the TWC fluctuation in

the inshore area. The northeasterly monsoon would greatly enhance the southwestward ZMCC, which

would then replace the northeastward TWC in the inshore area.

Figure 9

390

┌─────────────────────────────────────────────────────────────────┐
│                                                                 │
│                          Figure 10                              │
│                                                                 │
└─────────────────────────────────────────────────────────────────┘

392

┌─────────────────────────────────────────────────────────────────┐
│                                                                 │
│                          Figure 11                              │
│                                                                 │
└─────────────────────────────────────────────────────────────────┘

394

**3.3 Dynamic diagnostics**

The wintertime (January and February 2009) mean of the water column momentum balance (Fig. 12) is

used to show the overall distribution of the fundamental forces over the ECS shelf. The Coriolis force

(Fig. 12a) is mainly balanced by the total pressure (Fig. 12b) in both branches, indicating the dominant

role of geostrophic balance in the wintertime TWC. However, the wind-induced surface friction plays an

important role in the TWC, especially in the inshore area and the Taiwan Strait (Fig. 12c). The bottom

friction has an impact north of Taiwan and in the shallow Taiwan Strait, in particular when significant

Kuroshio intrusion enhances the bottom flow (Fig. 12d). The effects of advection and acceleration are

predominantly local indicated by mostly incoherent small scale distributions (Figs. 12e, 12f), so they can

be ignored when studying the large-scale current of the wintertime TWC.


┌─────────────────────────────────────────────────────────────────┐
│                                                                 │
│                          Figure 12                              │
│                                                                 │
└─────────────────────────────────────────────────────────────────┘


The variation of the driving forces at two representative points P1 and P2 were used to analyze the

dynamics of synoptic fluctuations north of Taiwan and in the inshore area. Regarding the results from

the EOF analysis, the three force terms, namely Coriolis, total pressure, and wind (Fig. 13), were selected

to investigate the effect of the TSC on the fluctuation north of Taiwan (Fig. 9a) and the effect of wind on
the fluctuation in the inshore area (Fig. 9b).

In the area north of Taiwan, the cross-shore fluctuations were induced by the TSC intrusion. The variation
of alongshore Coriolis force (Fig. 13a, black line) was much greater than the cross-shore Coriolis force
(Fig. 13b, black line), which means that the fluctuation north of Taiwan was mainly in the cross-shore
direction. The Coriolis force (Fig. 13a, black line) was mainly balanced by the total pressure (Fig. 13a,
blue line), which means the currents fluctuations north of Taiwan are dominated by geostrophic balance.
As mentioned in Sect. 3.2, the TWC fluctuation north of Taiwan was associated with the TSC rather than
with the Kuroshio Current. Therefore, in the shallow coastal area the TSC mainly caused variations in
the depth-independent barotropic pressure gradients, which further generated the cross-shore fluctuation.
The mechanism can be interpreted as follows. When a larger TSC intrusion occurred, the isobaric slope
tilted downward from south to north, generating a cross-shore current from the coastal area to the offshore
area. On the contrary, when the TSC intrusion was weak, the Kuroshio intrusion from offshore to inshore
dominated north of Taiwan.

Wind friction (Figs. 13c and 13d) was a fundamental factor in generating the fluctuations in the inshore
area. Although the geostrophic balance dominated in the inshore branch for most of the time, the
episodically strong winter monsoon had an important role in generating the TWC fluctuations. The
northwestward direction Coriolis force (Fig. 13c, black line) shows that the southwestward ZMCC
occurred on Jan. 12, Jan. 22, and Feb. 14, 2009 and was associated with a northeasterly wind (Fig. 13c,
red line). It indicates that strong northeasterly monsoon in winter can reduce or even stop the
northeastward TWC in the inshore area, causing the intermittency of the TWC inshore branch.

---------------------------------------------------------------------------------
Figure 13
---------------------------------------------------------------------------------


**4 Discussion**
Simulated results in the winters (December-March) of the years 2010 to 2013 (Fig. 14) show that general
structures of the TWC in the other winters were similar to that in winter 2009 (Fig. 5 and Fig. 9), which
indicates that the results from the winter 2009 can be regarded as representative for the winter situation.
The two TWC branches and the two areas of strong fluctuations were present in all winters from 2009 to
2013, although their strength showed a certain inter- annual variability in accordance with the changing
surface forcing and boundary fluxes.

---------------------------------------------------------------------------------
Figure 14
---------------------------------------------------------------------------------


The wintertime TWC, which is manifested by two subsurface branches and significant synoptic
fluctuations, has a very different structure when compared with the stationary and surface summertime
TWC reported in previous studies (Guan, 1978; Fang et al., 1991; Isobe, 2008). The synoptic events,
with time scales of 3-15 days, play a dominant role on the horizontal advective transports. According to
Ledwell et al. (1998) synoptic variations are much more effective on the horizontal transport than
variations on shorter time scales. The synoptic fluctuations modulate the spatial structure of the
wintertime TWC, especially when their magnitudes are comparable with that of the mean currents, such
as the two prominent fluctuations north of Taiwan and in the inshore area (Fig. 7). Therefore, the two
prominent fluctuations will be discussed next in terms of their contributions to the alongshore and cross-
shore transports.

**4.1 Cross-shore transport north of Taiwan induced by the TSC**
In the area north of Taiwan, the TSC intrusion generated strong fluctuations of the TWC in the cross-
shore direction (Fig. 9a). When a larger TSC intrusion occurred, the isobaric slope tilted downward from
south to north, generating a cross-shore current from the coastal area to the offshore area. Compared to
the reported summer route that transports Taiwan Strait water to the inshore area between the 30 and 100
m isobaths (Guan, 1978; Fang et al., 1991; Isobe, 2008; Yang et al., 2011, 2012), our results showed that
most Taiwan Strait water was transported to the TWC offshore branch and to the Kuroshio area as a result
of the cross-shore fluctuations induced by the synoptic TSC intrusion.

A numerical tracer simulation was used to analyze the role of the cross-shore fluctuation in the transport
of the TSC water and the Kuroshio water north of Taiwan. In order to demonstrate the characteristics of
the flow patterns more clearly, artificial tracers are released in the model domain and transported by the
velocity field provided by the FVCOM simulation. The tracer running was part of the FVCOM simulation;
therefore, all the above mentioned dynamics were involved, e.g., tide, wind, and boundary forces. The
release location and start date of the particles were configured as follows. Two sections, one in the Taiwan
Strait (Fig. 15a, black dots) and another in the East Taiwan Channel (Fig. 15b, black dots), were selected
as the source locations for the water masses of the TSC and the Kuroshio, respectively. The particles
were released on January 1, 2009 and tracked until March 31, 2009 (a total of 90 days).

Figure 15a shows the traces originating from the TSC area. Unlike the traditional route, where the TSC
water flows from the Taiwan Strait to the inshore area between the 30 and 100 m isobaths, most particles
(Fig. 15a, gray lines) were concentrated in the offshore branch under the effect of cross-shore fluctuation.
Two particles were selected to show the inshore route (Fig. 15a, red line) and offshore route (Fig. 15a,
blue line), with both passing the area north of Taiwan. When the two particles arrived at the area north
of Taiwan, the behavior of the tracers, according to specific velocity conditions (Fig. 15c), was very
different: a northwestward transport occurred on Jan. 25 for the inshore particles (Fig. 15c) and a
northeastward transport occurred on Feb. 12 for the offshore particles (Fig. 15c). The velocity conditions
in the area north of Taiwan corresponded to the variation of the Taiwan Strait flux (Fig. 10), which shows
that the Taiwan Strait flux on Feb. 12 was much greater than on Jan. 25. Therefore, it can be concluded
that the TSC intrusion induced an offshore transport north of Taiwan.

Figure 15b shows the traces originating from the Kuroshio area. In the same way as the TSC water, the
Kuroshio water was also transported to the northern shelf via both the inshore branch and the offshore
branch. The separation of the two branches north of Taiwan was caused by cross-shore fluctuations of
the currents. When the two particles arrived at the area north of Taiwan, a northwestward transport
occurred on Feb. 2 for the inshore particles (Fig. 15c) and a northeastward transport occurred on Feb. 12
for the offshore particles (Fig. 15c). This means that the offshore transport induced by the TSC also had
an effect on the distribution of Kuroshio water north of Taiwan. Liu et al. (2016) showed that the winter
TSC originated from a small branch of Kuroshio intrusion into the Luzon Strait. Our results complement
this picture, since they show that most TSC particles flow into the TWC offshore branch under the
influence of cross-shore fluctuation.

| Figure 15 |
|---|



Our results may underestimate the impact of Kuroshio intrusion on the fluctuation of the TWC northeast
of Taiwan, especially at the seasonal and interannual time scales. Wei et al. (2013) demonstrated that the
annual and interannual variations of the Kuroshio volume transport are large. In addition, Zhou et al.
(2015) pointed out that the annual and interannual variations of the Kuroshio intrusion northeast of
Taiwan are prominent. Liu et al. (2014b) presented supportive evidence that the Kuroshio intrusion, from
east of Taiwan to the onshore area north of Taiwan, is closely related to the Kuroshio volume transport.
This relation between the Kuroshio intrusion and the Kuroshio volume transport had been interpreted by
Su and Pan (1987) as the $\beta$-effect because of the sudden change in topography northeast of Taiwan. Our
results show that the intra-seasonal variation of the Kuroshio intrusion and the Kuroshio volume transport
was negligible compared with the TSC variation at the same time scale, indicating that the synoptic
fluctuation of TWC north of Taiwan is mainly induced by the TSC. However, because FVCOM uses
sigma co-ordinates in the vertical which are prone to errors in regions of steep topography, our results
may underestimate the fluctuations at the shelf break, in particular to the northeast of Taiwan where
Kuroshio intrusion occurs.

**4.2 Water exchange in the inshore area induced by wind**
In the inshore area, the synoptic fluctuations of the TWC (Fig. 9b) caused by wind were generally strong
in the alongshore direction and regionally important (along the latitudes 26.5 N and 28 N) in the cross-
shore direction. The alongshore fluctuations showed that the TWC inshore branch occurred episodically.
This episodic occurrence of the TWC agrees with the results from a previous study based on four mooring
surveys off the Zhe-Min coast (Zeng et al., 2012). The mechanism of the episodic occurrence of the TWC
was mainly associated with the winter monsoon, which agrees with the analysis of observational data by
Huang et al. (2016). However, the overall magnitude of the TWC fluctuation, and its role on the cross-
shore flux, are still not fully understood due to the short-term nature of the observational data.

We investigated the magnitude of TWC fluctuation, and its role on the water exchange, in the inshore
area. Previous studies (Su and Pan, 1987; Zeng et al., 2012) show that the TWC flows between the 50
and 100 m isobaths, whereas the ZMCC water dominates the coastal area west of the 50 m isobath in the
surface layer. As mentioned when discussing Figure 9d, the strongest TWC could reach the coastal area
as close as the 30 m isobath, being stronger than those reported in the literature. Moreover, the area with
large fluctuations spanned the area between the 30 and 100 m isobaths (Fig. 9b), indicating that water
between the 30 and 100 m isobaths may be either ZMCC or TWC water.

The episodic occurrence of the TWC inshore branch is directly related to the relative importance of the
southwestward ZMCC (Fig. 16, blue arrows) and the northeastward TWC (Fig. 16, red arrows). In this
paper, only wind-induced synoptic fluctuations are considered, not short-term extreme storm events.
When the winter monsoon (the northeasterly wind) prevails, the ZMCC occupies most of the inshore
area and the TWC inshore branch weakens (Fig. 16a). On the contrary, the TWC inshore branch can
intrude into the near-coast area under southwesterly wind conditions (Fig. 16b). The boundary between
the coastal current and the TWC may shift from the 100 m isobaths to the 30 m isobath in the cross-shore
direction, covering the entire area of the TWC inshore branch.

Our results further reveal that strong wind-induced cross-shore fluctuations occur in the inshore area (Fig.
9f). This cross-shore fluctuation has a significant ecological impact because of the connected nutrient
transport (Zhao and Guo, 2011). Ren et al. (2015) observed a cross-shore flux in the inshore area, which
was triggered by the transition of northeasterly to southwesterly winds. Their observed features can be
further interpreted with our result that wind-induced fluctuations can affect the cross-shore water
transport in the inshore area.

Largest cross-shore fluctuations were located at the latitudes 26.5 °N and 28 °N (Fig. 9f), which agreed
well with the latitudes where the TWC offshore meanders occurred in the mean currents (Fig. 5a). Thus
the offshore transports were most significant along the latitudes 26.5 °N and 28 °N according to both the
mean currents and the synoptic fluctuations. The offshore transport may be associated with the offshore-
penetrating fronts of coastal water in the ECS. Many remote-sensing images (He et al. 2010; Bai et al.
2013) have exhibited offshore-penetrating fronts that crossed the 70 m isobath and played an important
role in cross-shore material exchange, but the mechanisms of the offshore-penetrating fronts are still
under debate. Yuan and Qiao (2005) pointed out that both downwelling- and upwelling-favorable winds
are associated with the occurrence of the offshore-penetrating front. Ren et al. (2015) suggested that the
penetrating front is generated by the transition of northeasterly to southwesterly winds. Wu (2015)
suggested that the offshore-penetrating front is the response of buoyant coastal water to an along-isobath
undulation of the ambient pycnocline, which is controlled by a temperature stratification of the water
column. Our study offers a new interpretation, i.e., that the penetrating front is generated through the
wind-induced fluctuations and the TWC offshore meanders.


Figure 16


**5 Conclusions**
The FVCOM model was able to reproduce the wintertime TWC in 2009 reasonably well, as shown by a
validation in terms of the overall structure of the surface mean currents, the ECS boundary fluxes, and
data from four mooring stations. The validation showed that the simulated TWC was comparable to the
observed results, not only in terms of the mean currents but also in terms of the synoptic fluctuations.

The wintertime TWC showed two branches: one inshore and another offshore. The inshore branch
covered an area between the 30 and 100 m isobaths and flowed northeastward via a straight route. The
offshore branch was located between the 100 and 200 m isobaths and showed two prominent meanders.
It was shown that the Coriolis force was nearly balanced by the pressure gradient in both branches,
indicating the dominant role of the geostrophic balance for the mean current in both branches.

Two regions with significant synoptic fluctuations, north of Taiwan and the inshore area, were
investigated using the EOF method. The first two leading modes explained 91% of the total variance.
EOF1 showed that fluctuations occurred in the cross-shore direction south of 26 °N. These fluctuations
were mainly associated with variation of the TSC flux. EOF2 showed significant fluctuation between the
30 and 100 m isobaths. These fluctuations caused the episodic existence of the TWC inshore branch in
the alongshore direction and cross-shore fluctuations mainly at latitudes 26.5 °N and 28 °N, which were
mainly associated with the variation of wind speed.

We also studied the different dynamic reasons for the fluctuations in the two regions. In the area north of
Taiwan, the TSC and Kuroshio converged to initiate the TWC. A barotropic pressure anomaly was
generated by TSC intrusion from the Taiwan Strait causing a barotropic pressure gradient in the
alongshore direction; this explains why the synoptic fluctuations in this area occurred in the cross-shore
direction. Additionally, the wind had a strong effect on the synoptic fluctuations in the inshore area. The
northeasterly monsoon enhanced the southwestward ZMCC and replaced the TWC in the inshore area.
This situation is reversed during the southwesterly wind.

The synoptic fluctuations north of Taiwan and in the inshore area are important for both the alongshore
and cross-shore transports. Due to the fluctuation north of Taiwan, the mixed water of the TSC and the
Kuroshio was transported to both the inshore area and the offshore area, whereas most Taiwan Strait
water was transported to the offshore area in winter. The inshore fluctuation not only caused an episodic
occurrence of the TWC in the alongshore direction, which affected the alongshore transport of ZMCC
water and TWC water between the 30 and 100 m isobaths, but also impacted the cross-shore transports
along latitudes 26.5 °N and 28 °N.

**Acknowledgement**
The authors sincerely thank Dr. John M. Huthnance and the three anonymous reviewers for insightful
suggestions that improved this manuscript. This study was jointly supported by the Sino-German
cooperation in ocean and polar research under the grant BMBF-03F0701A (CLIFLUX), the National
Natural Science Foundation of China (U1609201, 41621064, 41306025), the grant from the scientific
research fund of the Second Institute of Oceanography, SOA (QNYC201603) and the project of State
Key Laboratory of Satellite Ocean Environment Dynamics, the Second Institute of Oceanography
(SOEDZZ1512).

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

**Table Captions**
Table 1: Annual-mean volume transports (Sv $= 10^6$ m$^3$/s) through various sections. The sections are
shown in Figure 2 using blue dashed lines.

**Figure Captions**

Figure 1: Density ($\sigma_t$, kg/m$^3$) distributions at 50 m depth derived from the GDEM climatological data in February (a), an ocean survey from Feb. 1–27, 2007 (b), and an ocean survey from Feb. 3–16, 2007 (c), with the density anomalies between the GDEM data and the two surveys (d and e). The two blue arrows indicate the two TWC branches in winter. The 30, 50, 70, 100 and 200 m isobaths are indicated with grey lines in panel a.

Figure 2: The FVCOM model grid (Left) and the surface mean flow in the ECS in winter (Right). The colors in the left panel show the grid length (km). The letters a, b, and c indicate the three open boundaries at the Taiwan Strait, the northwest Pacific Ocean, and the Japan/East Sea, respectively. The blue dashed lines (right) show some important straits around shelf boundary, including the Taiwan Strait (TWS), the East Taiwan Channel (ET), the Tsushima Strait (TUS), the Tokara Strait (TOS), and shelf break at the 200 m isobath. The red rectangle shows the study area of the wintertime TWC. The four red numbers off the Zhe-Min coast shows the four mooring sites observed from Jan. 5 to Feb. 28, 2009.

Figure 3: Validations of the wintertime TWC (warm color) along the section off the Zhe-Min coast (the short line with four red numbers in Figure 2): (a) observed alongshore currents; (b) simulated alongshore currents; (c) observed cross-shore currents; (d) simulated cross-shore currents. Note, an enlarged color scale is used for the cross-shore component to have a clear view of its weak structure.

Figure 4: Validations of the wintertime TWC fluctuations: (a) observed alongshore currents; (b) simulated alongshore currents; (c) observed cross-shore currents; (d) simulated cross-shore currents. The

observation data comes from Station 4 in Figure 1 and the simulated data has the same position and
period as the observation data.

Figure 5: a) Distribution of flow axes in the ECS in winter. The black arrows show the maximum velocity
(m/s) in the vertical profile (VMV) and the color shows the speed of the VMV. The two blue arrows with
label IB and OB represent the flow axes of the inshore branch and offshore branch, respectively. The red
line DL1 represents the dividing line between the coastal current and inshore branch, and the red line
DL2 separates the two TWC branches. b) Depth (m) of flow axes in the ECS, shown by color. Sections
S1–S6 were selected to study the wintertime TWC. c) Flux of inshore branch (blue) and offshore branch
(red) at different latitudes. Dashed lines show the positions of Sections. S1–S6. Note, the scale is not
linear.

Figure 6: Distributions of current speed along the six sections S1–S6 in winter. The blue arrow on the
left indicates the inshore branch according to the velocity cores from section S3 to S6. The blue arrow
on the right indicates the offshore branch according to the velocity cores from section S2 to S6. TSC is
the Taiwan Strait Warm Current.

Figure 7: Current standard deviation in the layer of the VMV. The black arrows indicate the major axis
of the ellipse which represent the standard deviation of the current. The color shading shows the
respective magnitude. The two blue arrows indicate the two TWC branches. The red curve indicate the
area where the current standard deviation is larger than 0.1 m/s and the branches' representative points
(P1 and P2) are selected for later analysis.

Figure 8: Variation of alongshore currents (m/s, shown by color scale) for the entire water column north
of Taiwan (P1) and in the inshore area (P2) and their relation with upper mixed layer depth. The positive
velocity (warm color) indicates the occurrence of the TWC. The gray solid lines show the depth of the
upper mixed layer.

Figure 9: The spatial pattern of the first (EOF1; left) and second (EOF2; right) leading modes of the
VMV in the ECS: (a) EOF1 currents, (b) EOF2 currents, (c) EOF1 alongshore component, (d) EOF2
alongshore component, (e) EOF1 cross-shore component, and (f) EOF2 cross-shore component (all
shown by black arrows with the color representing the magnitude). The 30, 50, 70, 100 and 200 m
isobaths are indicated with grey lines.

Figure 10: Temporal variation of EOF1, north-south component of wind speed, surface net heat flux, and
TSC flux across the TWS section, and Kuroshio flux across the ET section. Their linear correlation
coefficients R and time-lags are also indicated in each panel. The p value is a declining indicator which
indicates the impact significance of the linear correlation coefficients R whereby R has statistical
significance and the confidence level is larger than 95% when the p value is less than 0.05.

Figure 11: Temporal variation of EOF2, north-south component of wind speed, surface net heat flux, and
TSC flux across the TWS section, and Kuroshio flux across the ET section. Their linear correlation
coefficients and time-lags are also indicated in each panel.

Figure 12: The effects of Coriolis force (a), total pressure (b), surface friction (c), bottom friction (d),
advection (e), and local acceleration (f) for water column in winter according to Eq. (5) (shown by black
arrows with the color representing the magnitude; units: $10^{-4}$ $m^2/s^2$). The two blue arrows indicate the
two TWC branches. The two triangles indicate the two regions with significant fluctuation north of
Taiwan (P1) and in the inshore area (P2).

Figure 13: Variations in Coriolis force, total pressure, and wind in the cross-shore direction at P1 (a), the
alongshore direction at P1 (b), the cross-shore direction at P2 (c), and the alongshore direction at P2 (d)
according to Eq. (5). The grey pointers indicate the alongshore and cross-shore directions of dynamical
effects in the earth coordinate system.

Figure 14: Mean currents (upper panels) and synoptic fluctuations (EOF1 in middle panels and EOF2 in
bottom panels) in winters of 2010-2013. The black arrows in the upper panels show the velocity (m/s) in
the layer of VMV with the color representing the current speed. The two blue arrows with label IB and
OB represent the flow axes of the inshore branch and offshore branch, respectively. The black arrows in
the middle panels and bottom panels represent the EOF components (m/s) with their magnitude
represented by color scales.

Figure 15: Traces of TSC water (a) and Kuroshio water (b) in winter, with the variation of surface currents
north of Taiwan (c). The green lines L1 and L2 indicate the starting latitude of the tracers (24.5 °N) and
the latitude which is representative for synoptic fluctuations north of Taiwan (25.8 °N), respectively. The
black dots represent the release locations of tracers originated from line L1. The gray lines show the
entire trajectories of the tracers. The red lines and blue lines are selected trajectories, which are close to
the inshore branch and offshore branch, respectively. The dates show the times when selected tracers
cross the latitude indicated by line L2. The numbers are the depths of the tracers, which are labeled at an
interval of six days. The two black arrows represent the two TWC branches.

Figure 16: The VMV under the northerly wind (a) and southerly wind (b). Panel (c) shows the variation
of wind in winter. Blue vectors and red vectors show the southwestward coastal current and the
northeastward TWC, respectively. Gray contours indicate the 30, 50, 70, and 100 m isobaths. The two
black arrows represent the two TWC branches. The green ellipse indicates the inshore area with
significant fluctuation.

Table 1: Annual-mean volume transports (Sv = $10^6$ m³/s) through various sections. The sections are shown in Figure 2 using blue dashed lines.

| Section | Present model | Previous estimates |
| --- | --- | --- |
| **Taiwan Strait** | 1.22 | 1.2 (Isobe, 2008) |
| | | 1.8 (Wang et al., 2003) |
| | | 1.09 (Wu and hsin, 2005) |
| | | 1.03 (Yang et al., 2011) |
| | | 1.72 (Guo et al., 2006) |
| | | 0.5 (Hung et al., 2003) |
| | | 1.10 (Liu et al., 2014b) |
| **Tsushima Strait** | 2.85 | 2.65 (Isobe, 2008) |
| | | 3.03 (Guo et al., 2006) |
| | | 2.70 (Yang et al., 2011) |
| | | 2.52 (Liu et al., 2014b) |
| **200m isobath** | 1.66 | 1.46 (Guo et al., 2006) |
| | | 0.87 (Liu et al., 2014a) |
| | | 3.0 (Teague et al., 2003) |
| | | 2.74 (Lee and Matsuno, 2007) |
| **East Taiwan Channel** | 22.71 | 21.50 (Johns et al., 2001) |
| | | 23.00 (Teague et al., 2003) |
| | | 23.83 (Guo et al., 2006) |
| | | 28.4 (Hsin et al., 2013) |
| | | 21.37 (Yang et al., 2011) |
| | | 20.74 (Liu et al., 2014b) |
| **Tokara Strait** | 23.20 | 23.4 (Feng et al., 2000) |
| | | 20.00 (Teague et al., 2003) |
| | | 20.66 (Yang et al., 2011) |
| | | 24.42 (Liu et al., 2014b) |

878

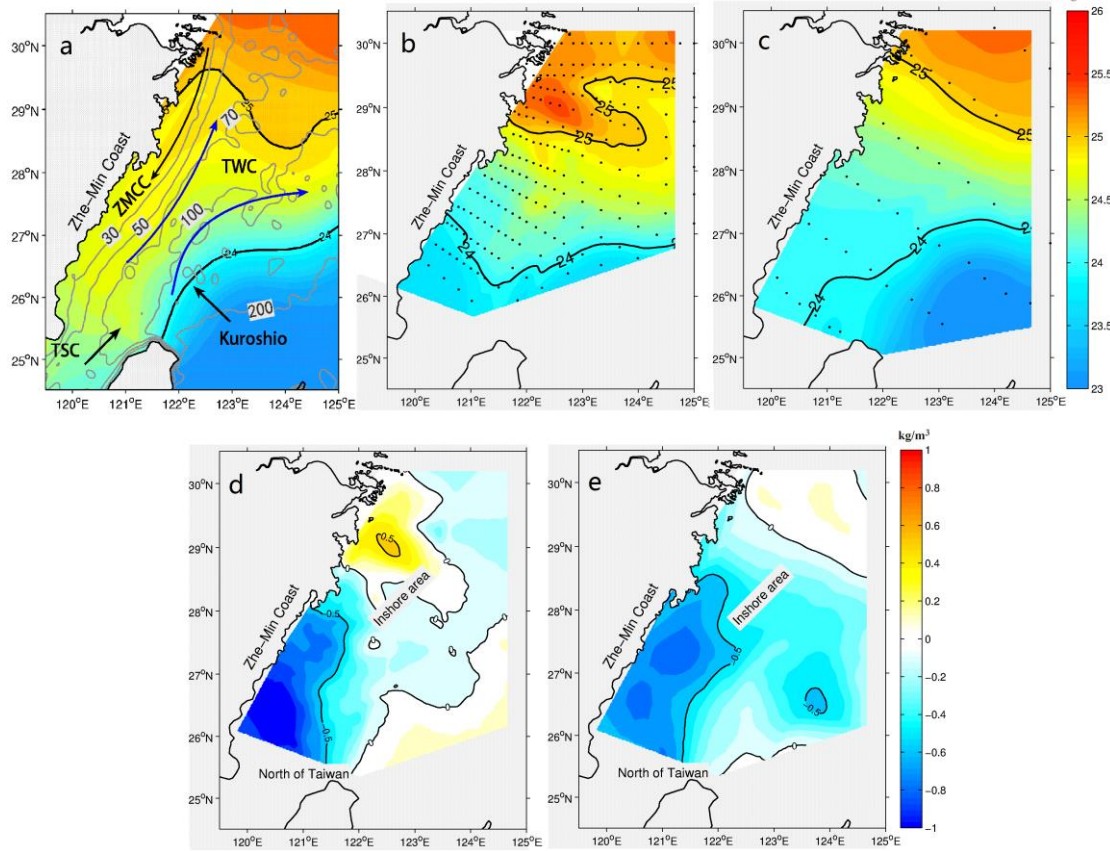

879

Figure 1: Density ($\sigma_t$, kg/m³) distributions at 50 m depth derived from the GDEM climatological data in

February (a), an ocean survey from Feb. 1–27, 2007 (b), and an ocean survey from Feb. 3–16, 2007 (c),

with the density anomalies between the GDEM data and the two surveys (d and e). The two blue arrows

indicate the two TWC branches in winter. The 30, 50, 70, 100 and 200 m isobaths are indicated with grey

lines in panel a.


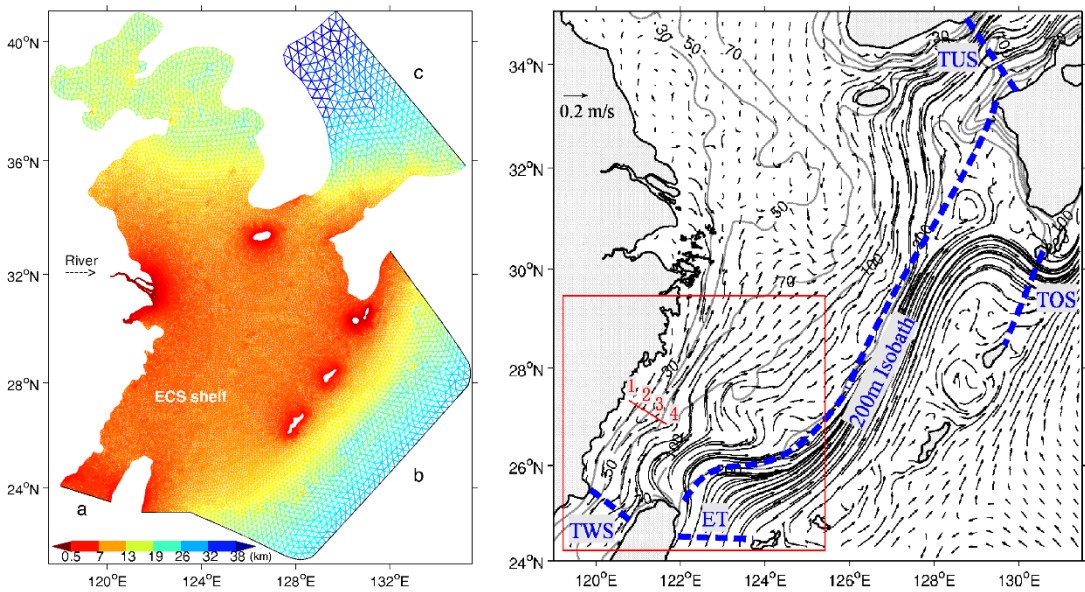


Figure 2: The FVCOM model grid (Left) and the surface mean flow in the ECS in winter (Right). The
colors in the left panel show the grid length (km). The letters a, b, and c indicate the three open boundaries
at the Taiwan Strait, the northwest Pacific Ocean, and the Japan/East Sea, respectively. The blue dashed
lines (right) show some important straits around shelf boundary, including the Taiwan Strait (TWS), the
East Taiwan Channel (ET), the Tsushima Strait (TUS), the Tokara Strait (TOS), and shelf break at the
200 m isobath. The red rectangle shows the study area of the wintertime TWC. The four red numbers off
the Zhe-Min coast shows the four mooring sites observed from Jan. 5 to Feb. 28, 2009.

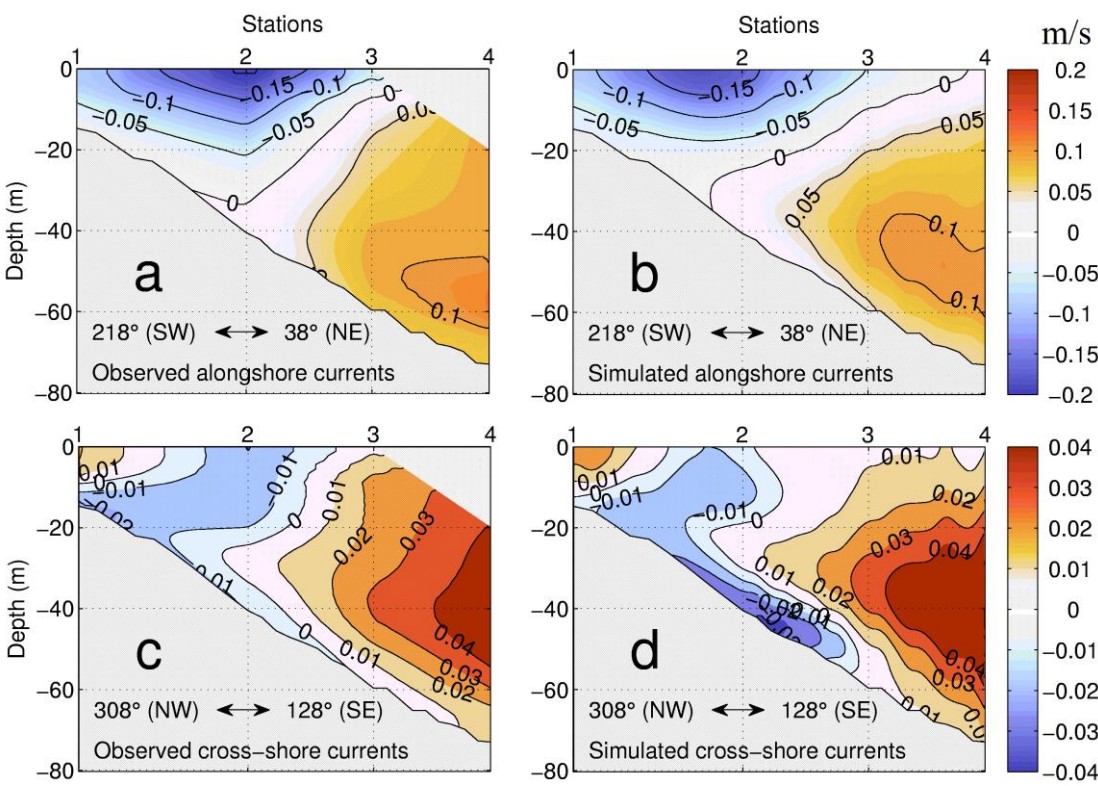


Figure 3: Validations of the wintertime TWC (warm color) along the section off the Zhe-Min coast (the
short line with four red numbers in Figure 2): (a) observed alongshore currents; (b) simulated alongshore
currents; (c) observed cross-shore currents; (d) simulated cross-shore currents. Note, an enlarged color
scale is used for the cross-shore component to have a clear view of its weak structure.

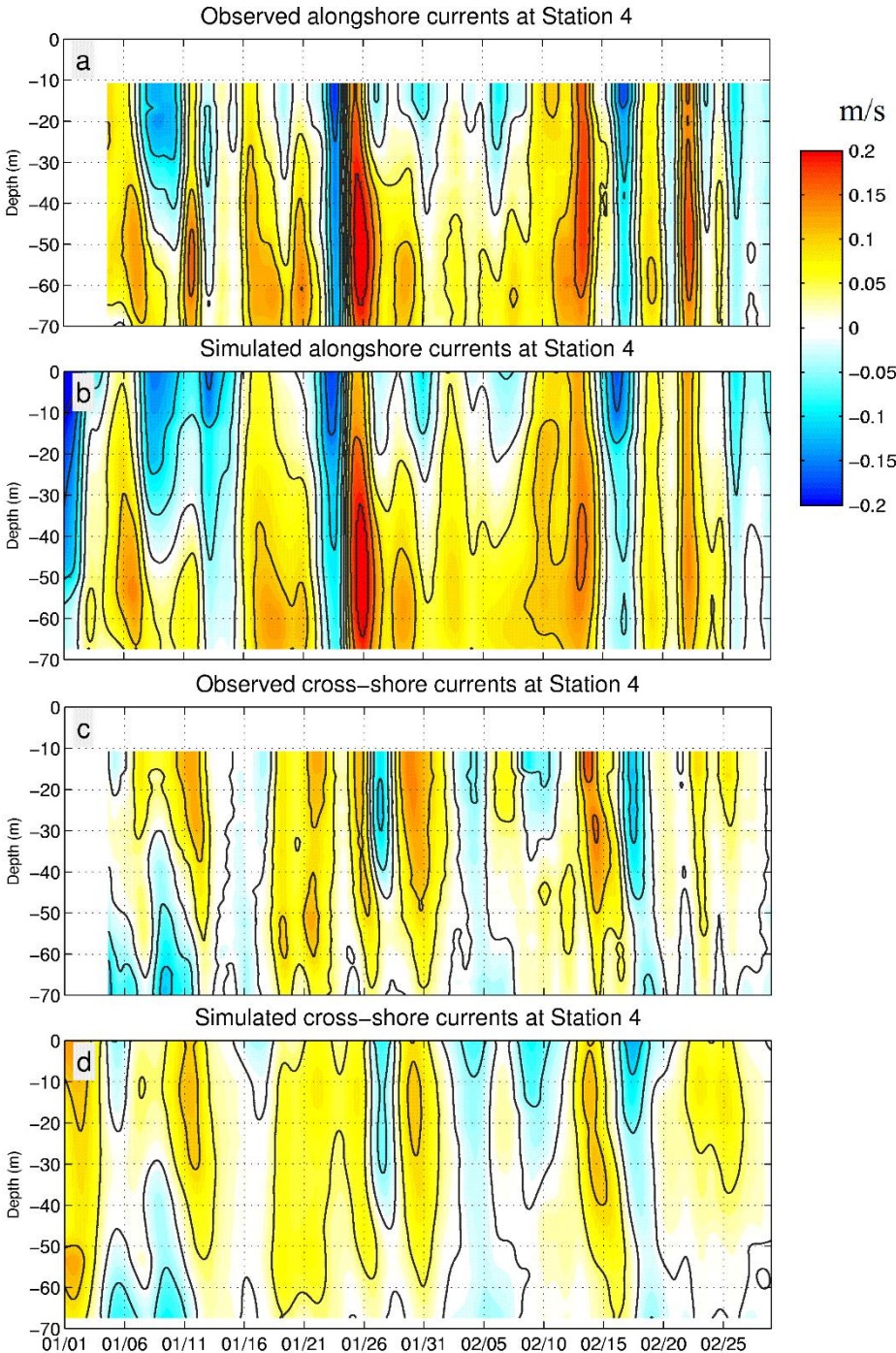


Figure 4: Variations of the inshore branch of TWC during January and February 2009: (a) observed
alongshore currents; (b) simulated alongshore currents; (c) observed cross-shore currents; (d) simulated
cross-shore currents. The observation data comes from Station 4 in Figure 1 and the simulated data has
the same position and period as the observation data.

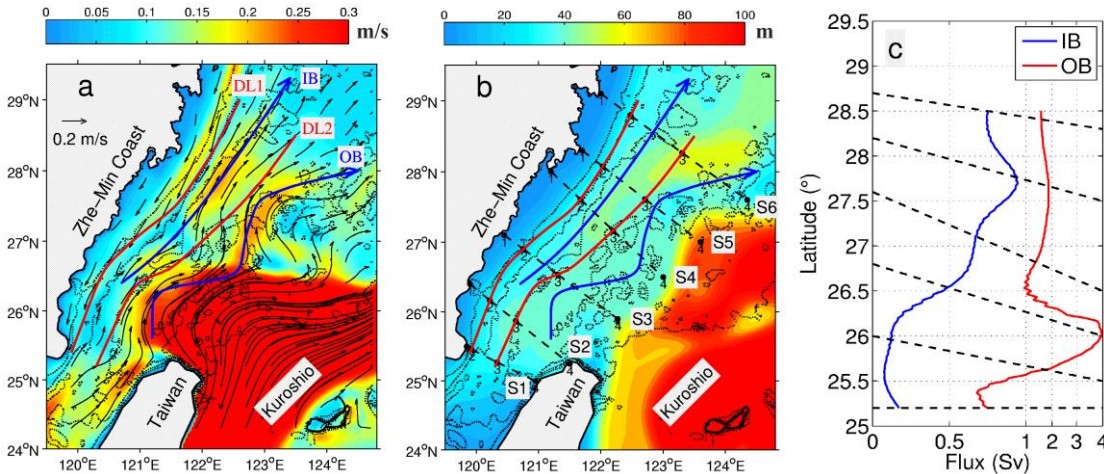


Figure 5: a) Distribution of flow axes in the ECS in winter. The black arrows show the maximum velocity
(m/s) in the vertical profile (VMV) and the color shows the speed of the VMV. The two blue arrows with
label IB and OB represent the flow axes of the inshore branch and offshore branch, respectively. The red
line DL1 represents the dividing line between the coastal current and inshore branch, and the red line
DL2 separates the two TWC branches. b) Depth (m) of flow axes in the ECS, shown by color. Sections
S1–S6 were selected to study the wintertime TWC. c) Flux of inshore branch (blue) and offshore branch
(red) at different latitudes. Dashed lines show the positions of Sections. S1–S6. Note, the scale is not
linear.

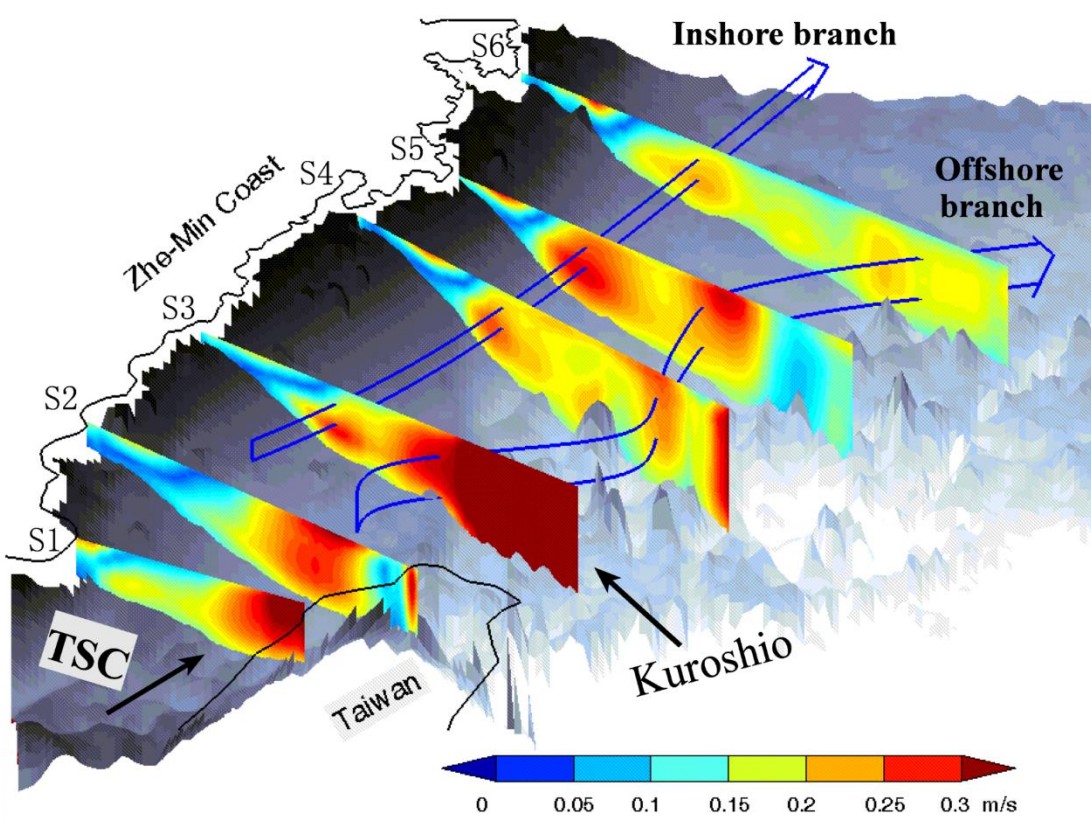


Figure 6: Distributions of current speed along the six sections S1–S6 in winter. The blue arrow on the
left indicates the inshore branch according to the velocity cores from section S3 to S6. The blue arrow
on the right indicates the offshore branch according to the velocity cores from section S2 to S6. TSC is
the Taiwan Strait Current.

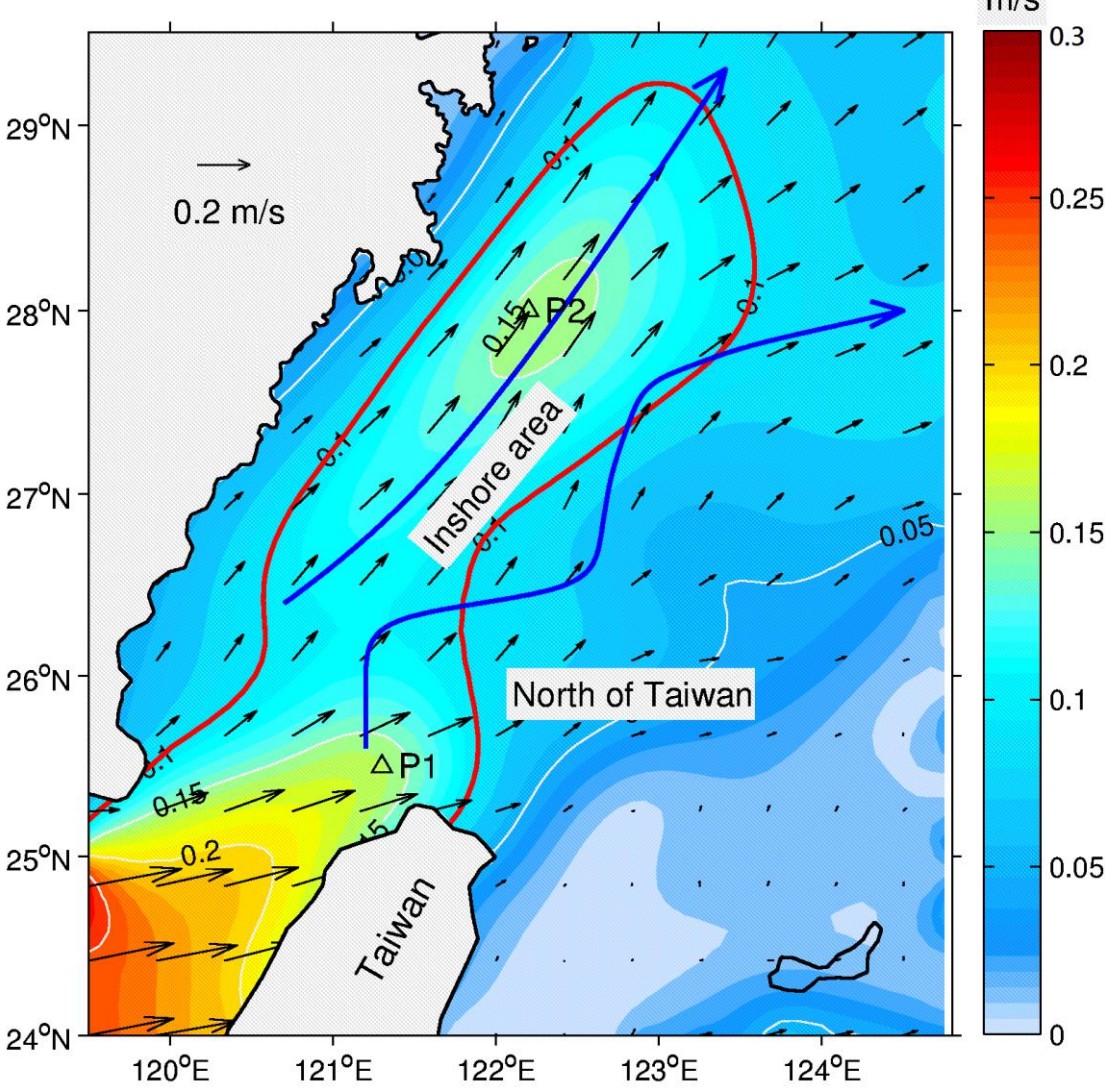


Figure 7: Current standard deviation in the layer of the VMV. The black arrows indicate the major axis

of the ellipse which represent the standard deviation of the current. The color shading shows the

respective magnitude. The two blue arrows indicate the two TWC branches. The red curve indicate the

area where the current standard deviation is larger than 0.1 m/s and the branches' representative points

(P1 and P2) are selected for later analysis.


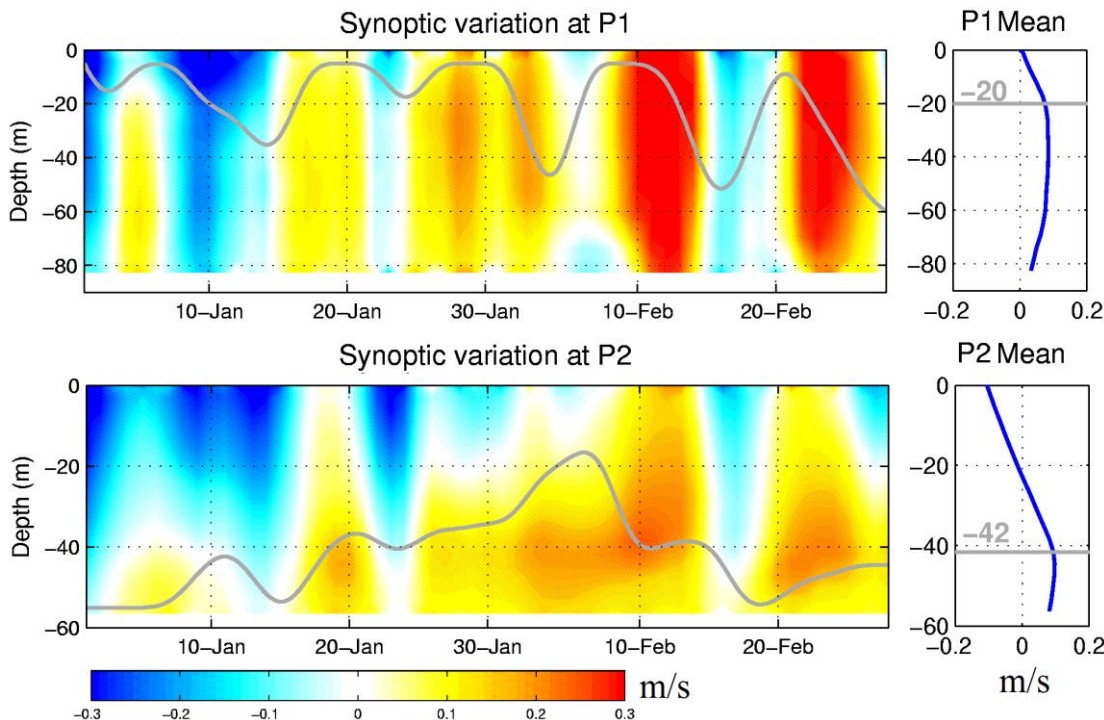


Figure 8: Variation of alongshore currents (m/s, shown by color scale) for the entire water column north
of Taiwan (P1) and in the inshore area (P2) and their relation with upper mixed layer depth. The positive
velocity (warm color) indicates the occurrence of the TWC. The gray solid lines show the depth of the
upper mixed layer.

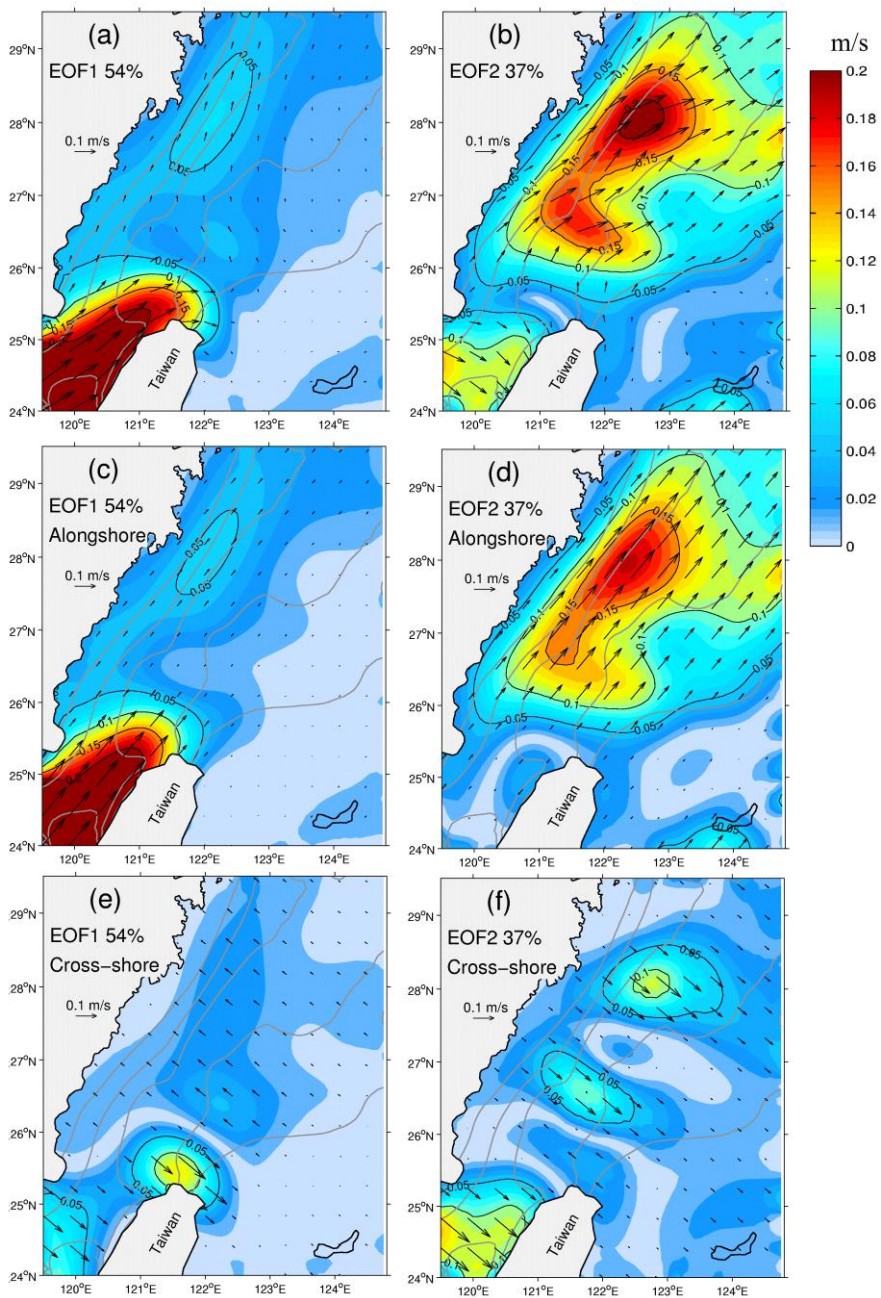


Figure 9: The spatial pattern of the first (EOF1; left) and second (EOF2; right) leading modes of the

VMV in the ECS: (a) EOF1 currents, (b) EOF2 currents, (c) EOF1 alongshore component, (d) EOF2

alongshore component, (e) EOF1 cross-shore component, and (f) EOF2 cross-shore component (all

shown by black arrows with the color representing the magnitude). The 30, 50, 70, 100 and 200 m

isobaths are indicated with grey lines.

942

943

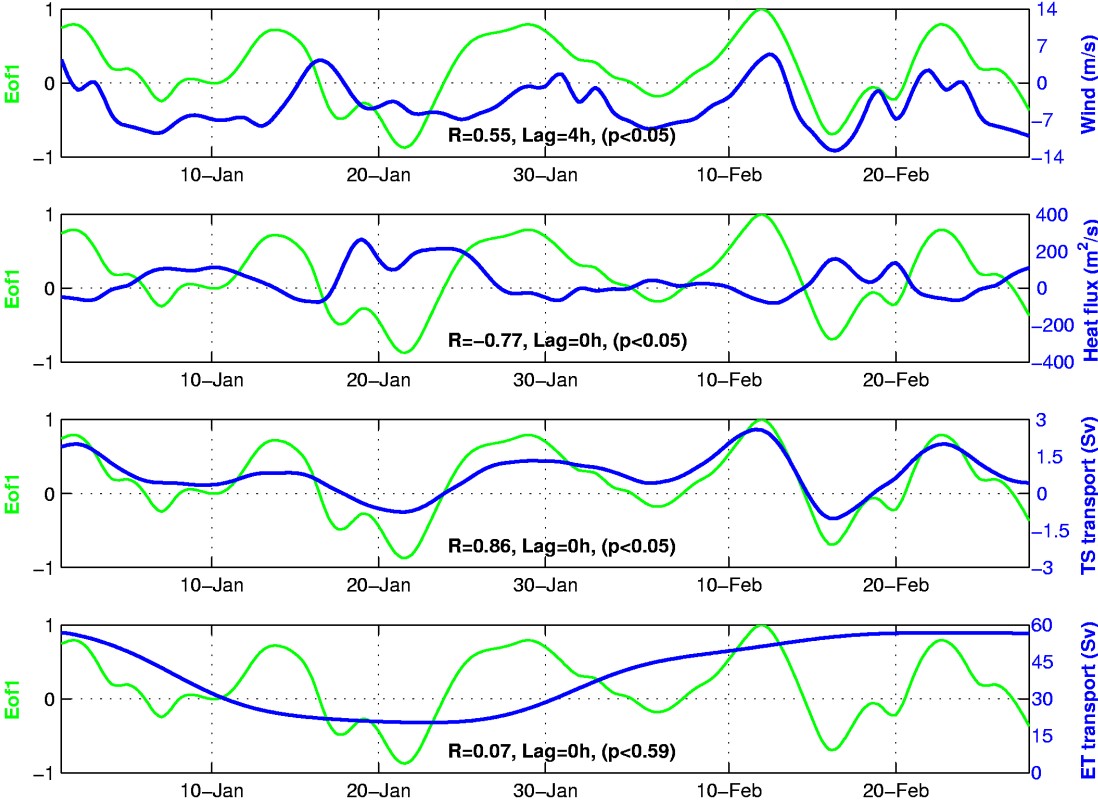

Figure 10: Temporal variation of EOF1, north-south component of wind speed, surface net heat flux, and

TSC flux across the TWS section, and Kuroshio flux across the ET section. Their linear correlation

coefficients R and time-lags are also indicated in each panel. The p value is a declining indicator which

indicates the impact significance of the linear correlation coefficients R whereby R has statistical

significance and the confidence level is larger than 95% when the p value is less than 0.05.


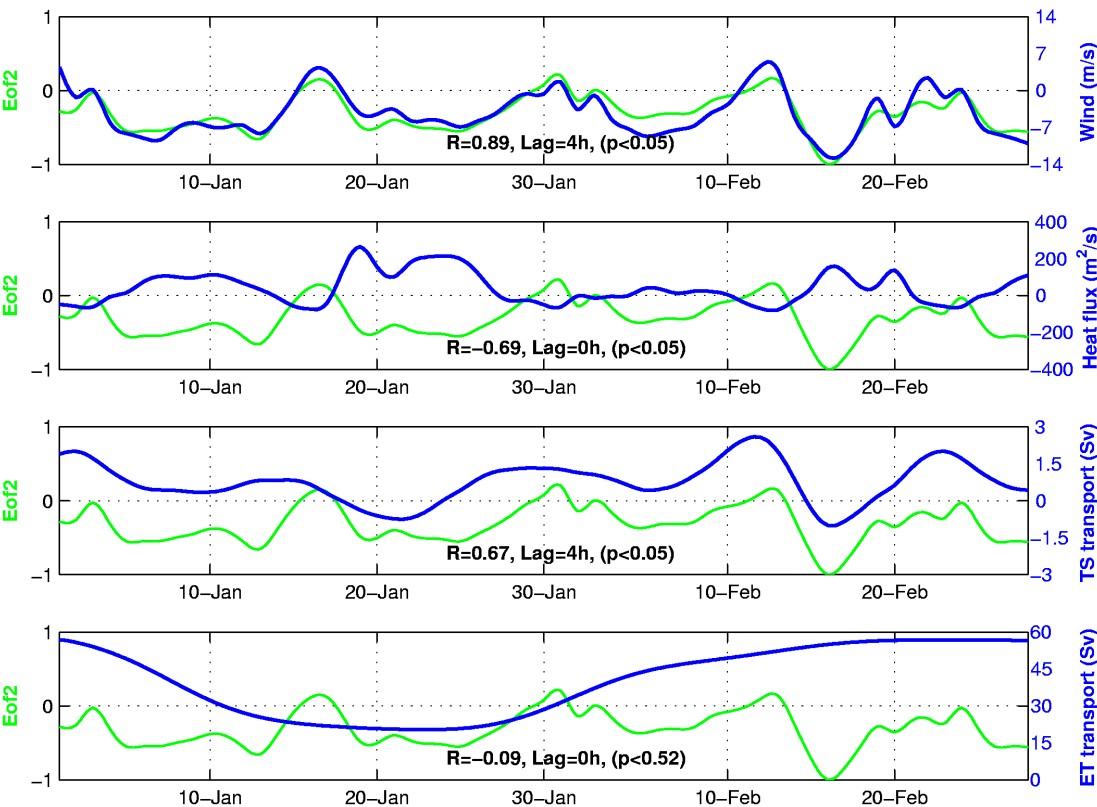


Figure 11: Temporal variation of EOF2, north-south component of wind speed, surface net heat flux, and
TSC flux across the TWS section, and Kuroshio flux across the ET section. Their linear correlation
coefficients and time-lags are also indicated in each panel.

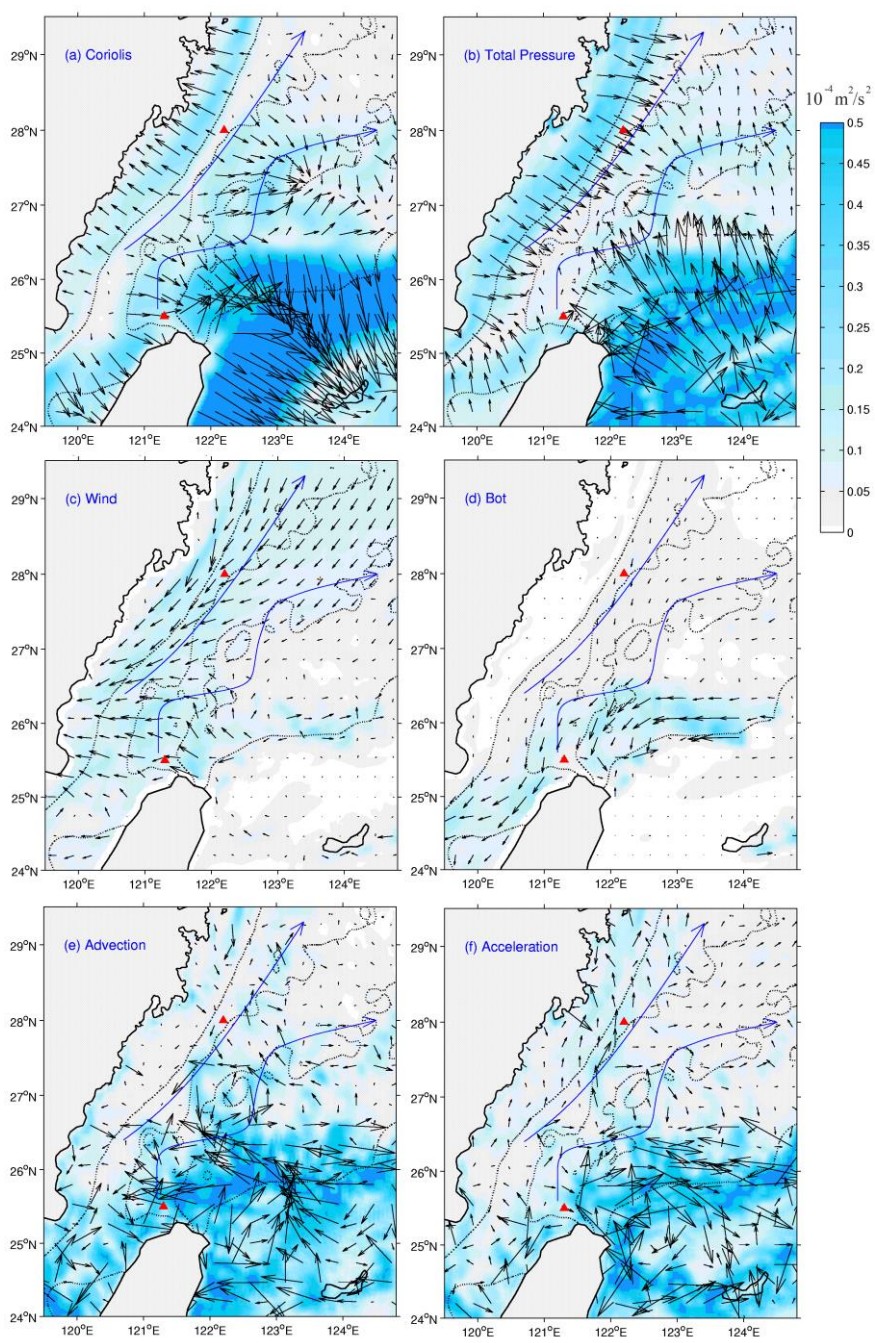


Figure 12: The effects of Coriolis force (a), total pressure (b), surface friction (c), bottom friction (d),

advection (e), and local acceleration (f) for water column in winter according to Eq. (5) (shown by black

arrows with the color representing the magnitude; units: $10^{-4}$ m$^2$/s$^2$). The two blue arrows indicate the

two TWC branches. The two triangles indicate the two regions with significant fluctuation north of

Taiwan (P1) and in the inshore area (P2).

961

962

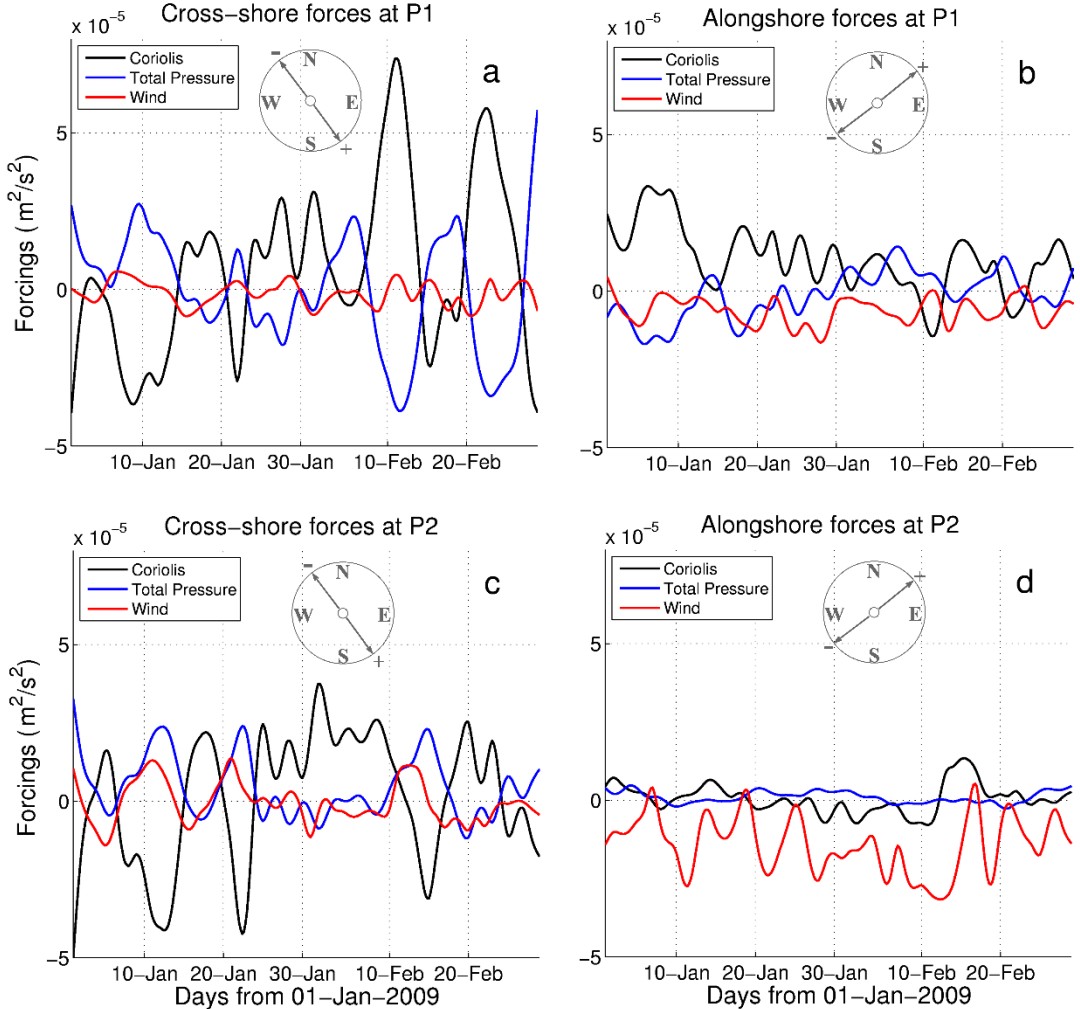

963

Figure 13: Variations in Coriolis force, total pressure, and wind in the cross-shore direction at P1 (a), the

alongshore direction at P1 (b), the cross-shore direction at P2 (c), and the alongshore direction at P2 (d)

according to Eq. (5). The grey pointers indicate the alongshore and cross-shore directions of dynamical

effects in the earth coordinate system.


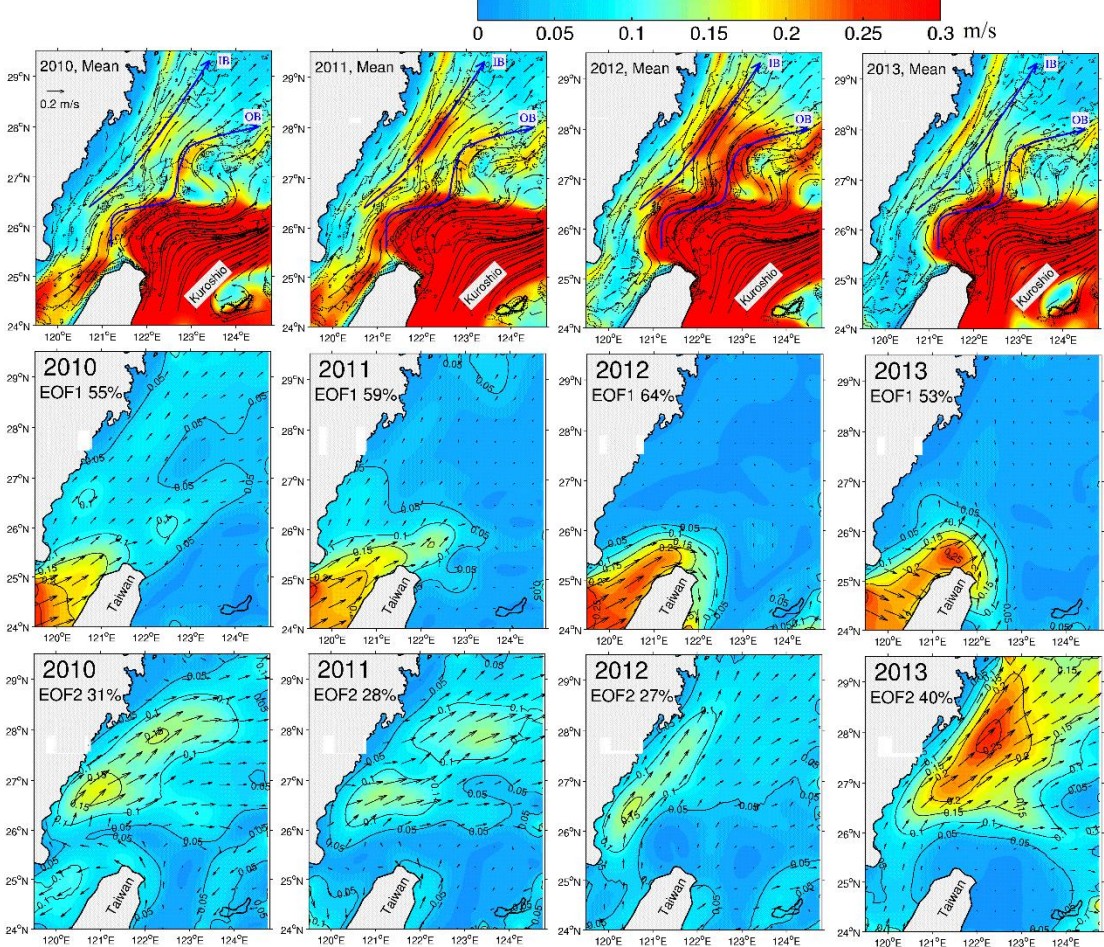


Figure 14: Mean currents (upper panels) and synoptic fluctuations (EOF1 in middle panels and EOF2 in
bottom panels) in winters of 2010-2013. The black arrows in the upper panels show the velocity (m/s) in
the layer of VMV with the color representing the current speed. The two blue arrows with label IB and
OB represent the flow axes of the inshore branch and offshore branch, respectively. The black arrows in
the middle panels and bottom panels represent the EOF components (m/s) with their magnitude
represented by color scales.

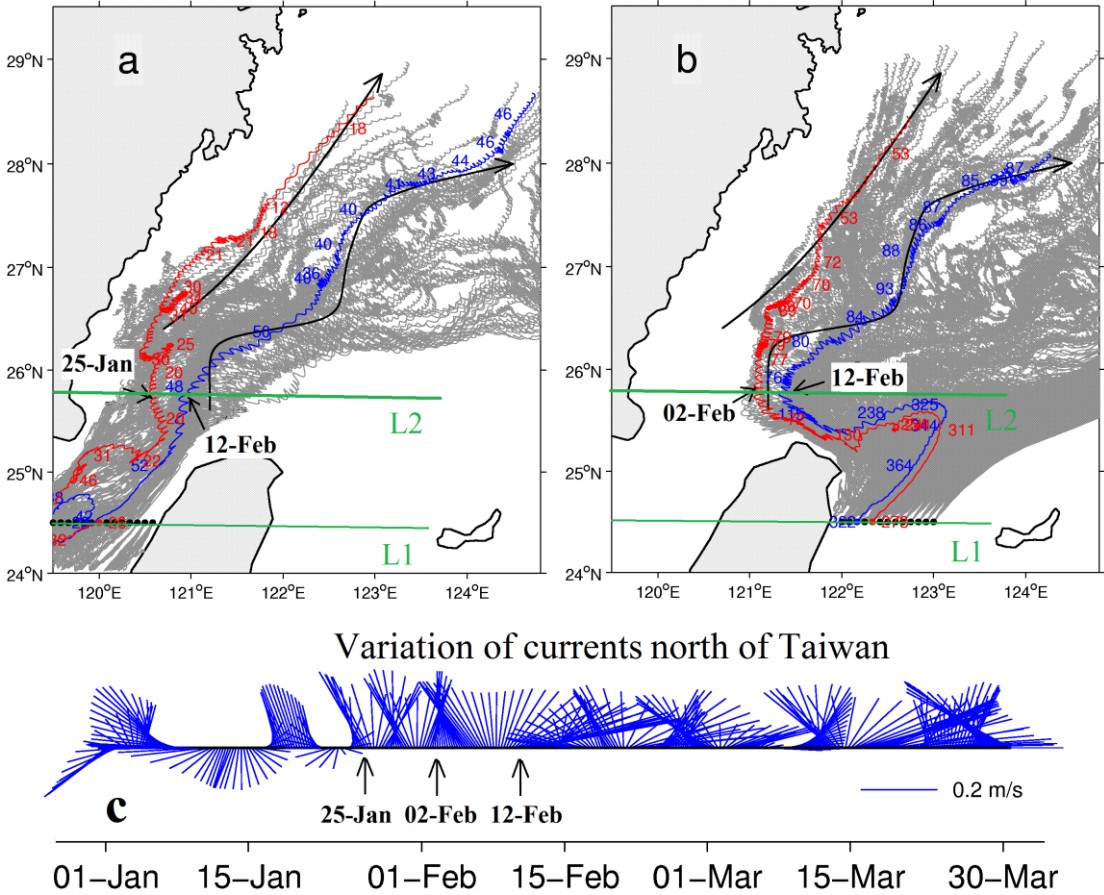


Figure 15: Traces of TSC water (a) and Kuroshio water (b) in winter, with the variation of surface currents
north of Taiwan (c). The green lines L1 and L2 indicate the starting latitude of the tracers (24.5 °N) and
the latitude which is representative for synoptic fluctuations north of Taiwan (25.8 °N), respectively. The
black dots represent the release locations of tracers originated from line L1. The gray lines show the
entire trajectories of the tracers. The red lines and blue lines are selected trajectories, which are close to
the inshore branch and offshore branch, respectively. The dates show the times when selected tracers
cross the latitude indicated by line L2. The numbers are the depths of the tracers, which are labeled at an
interval of six days. The two black arrows represent the two TWC branches.

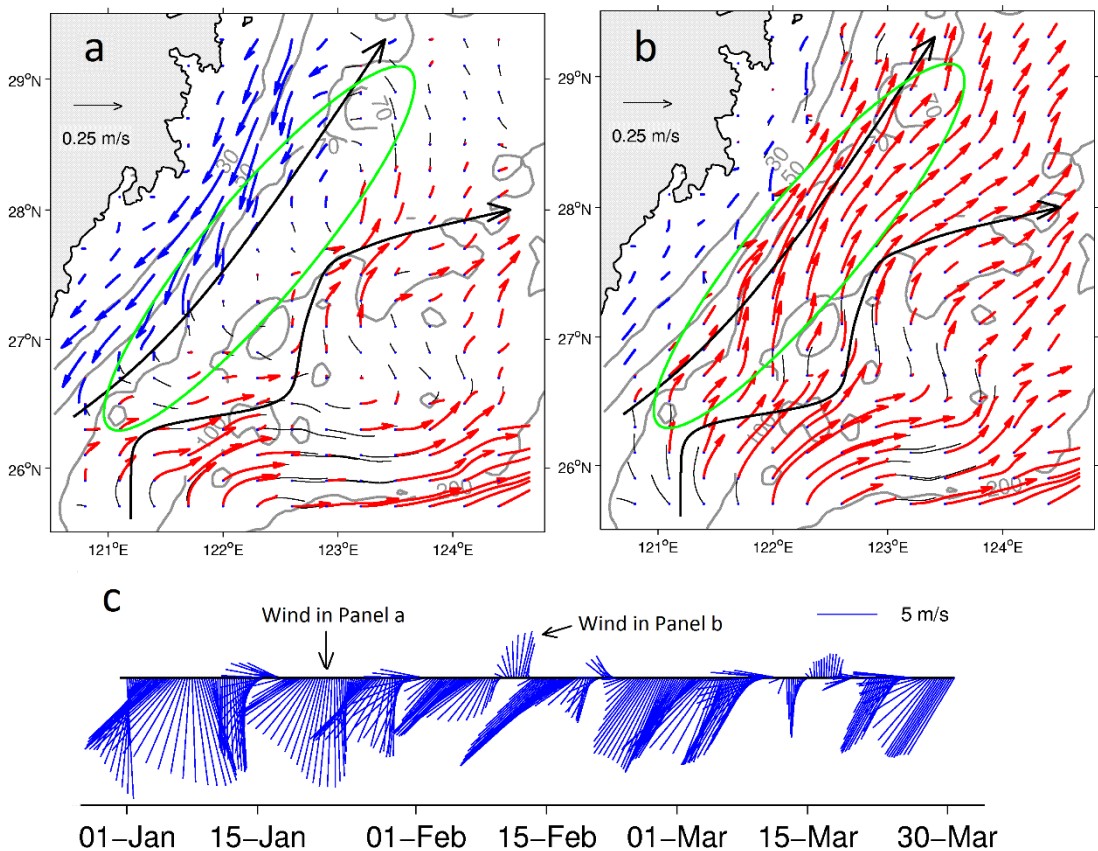


Figure 16: The VMV under the northeasterly wind (a) and southwesterly wind (b). Panel (c) shows the
variation of wind in winter. Blue vectors and red vectors show the southwestward coastal current and the
northeastward TWC, respectively. Gray contours indicate the 30, 50, 70, and 100 m isobaths. The two
black arrows represent the two TWC branches. The green ellipse indicates the inshore area with
significant fluctuation.
