# Peer review of "Synoptic fluctuation of the Taiwan Warm Current in winter on the East China Sea shelf"

_Ocean Science, 2016_

## Referee Comment (RC1) · Anonymous Referee #1 · 28 Sep 2016

In this paper, The seasonal mean and synoptic fluctuation of the wintertime Taiwan Warm Current (TWC) were investigated using a FVCOM model. It is found that two areas with significant fluctuations of the TWC were identified during wintertime. One fluctuation is related to the intrusion of the Taiwan Strait Current (TSC), the other is due to the wind effects. I agree that the conclusions of the paper are new and interesting, I recommend its publication after some minor revisions.

The minor comments are, 1. There are too many keywords, I suggest that "Taiwan Warm Current, Taiwan Strait Current, Kuroshio, Zhe-Min Coastal Current, East China Sea" are enough. 2. The model domain and open boundaries are not clear, they may be drawn or indicated in figure 1 or 2. 3. In figures 10&11, can you calculate the confidence level of the correlation? 4. A new paper by Liu et al. (2016, Numerical simulation of the Kuroshio intrusion into the South China Sea by a passive tracer. Acta

Oceanologica Sinica, 2016, 35(9), 1-12. ) used tracer simulation to study the Kuroshio intrusion, their model domain covered some of yours. I suggest in discussion of your Fig.14b, comparison between both model results is beneficial.
* * *

---

## Referee Comment (RC2) · Anonymous Referee #1 · 1 Oct 2016

My previous comments are well addressed, I'm satisfied with the authors' revised ms., I thus recommend its publication.

---

## Author Comment (AC1) · 1 Oct 2016

Dear Referee,

Thank you very much for your comments on our joint manuscript. According to these comments, we have made corrections which hopefully could clarify the points brought up by you. We responded to these comments one by one in the attachment. The revised manuscript is also attached in the supplement.

Best regards,

Jiliang Xuan, Daji Huang, Thomas Pohlmann, Jian Su, Bernhard Mayer, Ruibin Ding, Feng Zhou September 29, 2016

Response to referee

[Figure]

Comments: The minor comments are, 1. There are too many keywords, I suggest that "Taiwan Warm Current, Taiwan Strait Current, Kuroshio, Zhe-Min Coastal Current, East China Sea" are enough.

Author's response: We agree that seven keywords are too many. Therefore, we keep the first five keywords which are more important and more frequently used in the manuscript than the last two keywords.

Author's changes in manuscript: Line 41-43: The last two keywords "Zhe-Min Coastal Current, wind" were deleted.

Comments: 2. The model domain and open boundaries are not clear, they may be drawn or indicated in figure 1 or 2.

Author's response: We now have drawn the model domain and the open boundaries in Fig. 2 (Left panel).

Author's changes in manuscript: Line 801-809: We added a left panel in Fig.2 to show model domain, open boundaries and grids. The Fig. 2 will be attached in this reply.

Comments: 3. In figures 1011, can you calculate the confidence level of the correlation?

Author's response: We used the MATLAB function "corr" to compute the declining indicator, i.e., the p value which can directly be related to the confidence level. E.g., if the confidence level is larger than 95

Author's changes in manuscript: Line 856-857: We changed the statement "whereby R has statistical significance when the p value is less than 0.05." to "whereby R has statistical significance and the confidence level is larger than 95

Comments: 4. A new paper by Liu et al. (2016, Numerical simulation of the Kuroshio intrusion into the South China Sea by a passive tracer. Acta Oceanologica Sinica, 2016, 35(9), 1-12. ) used tracer simulation to study the Kuroshio intrusion, their model

domain covered some of yours. I suggest in discussion of your Fig.14b, comparison between both model results is beneficial.

Author's response: Thanks for the information of the new paper by Liu et al. (2016). We have carefully compared the MITgcm results from Liu et al. (2016) and the FVCOM results from this manuscript. Their results nice complement our study, since they provide the information on the origin of the TSC, which is missing in our paper. Liu et al. (2016) showed that the winter TSC originated in a small branch of Kuroshio intrusion from the Luzon Strait. Their Fig. 14 shows that the winter TSC will flow into both the ECS shelf and the Kuroshio region showing significant synoptic fluctuations.

Author's changes in manuscript: Line 455-458: We referred to "Liu et al. (2016)" in the text and discussed about the route of TSC from the two models. A discussion was added after "… the TSC also had an effect on the distribution of Kuroshio water north of Taiwan.": "Liu et al. (2016) showed that the winter TSC originated from a small branch of Kuroshio intrusion into the Luzon Strait. Our results complement this picture, since they show that most TSC particles flow into the TWC offshore branch under the influence of cross-shore fluctuation."

Please also note the supplement to this comment:
http://www.ocean-sci-discuss.net/os-2016-70/os-2016-70-AC1-supplement.pdf

**Supplement:**

[revised manuscript text omitted]

---

## Referee Comment (RC3) · Anonymous Referee #2 · 16 Oct 2016

Comments on 'Synoptic fluctuation of the Taiwan Warm Current in winter on the East China Shelf' by Xuan et al

In this manuscript the authors investigated short term variations (periods of 3-15 days) of the Taiwan Warm Current in the winter 2009 on the East China Shelf, using the results of multi-year model simulations of FVCOM. They identified two areas of strong fluctuations, namely, the north of Taiwan and the further northern area of the inshore region, and found that the main driving forces for the fluctuations were the Taiwan Strait Current and the winter monsoon winds. I find the paper is interesting and well written, merits to be published in the Ocean Science after some minor clarifications and corrections outlined below.

The title '. . ... in winter on the East China Sea Shelf'. In fact only results from the winter

[Figure]

(Feb) 2009 were analyzed and presented in the paper and no attempt was made to extend to other winters. I, therefore, suggest that the title to be changed to '. . .in the winter of 2009. . .', or similar.

Line 133-135, on the model, 'the river discharge of the Changjiang and Huanghe . . ..'. In the immediate region of the study, there are other important rivers, e.g., Qiantang, etc.; did the authors include them; if not, why not.

Validation of the model. The model results were validated in the coastal regional of the East China Sea shelf and good agreements between measurements and model were obtained. However this is limited to the shallow coastal region. It is well known that the FVCOM uses a sigma co-ordinate and prone to errors in the region of steep topography. Therefore authors should at least caution readers that in the slope region, such as the Kuroshio intrusion (line 465-470), model results are less reliable unless a good validation is provided.

Line 264. '. . . in wintertime, both branches flowed on the isobaths, which is fully in accordance with the conservation of potential vorticity'. It implies 'cross isobaths flow is not following the pv conservation law', which is not correct. btw, 'on the isobaths' > 'along the isobaths'.

---

## Author Comment (AC2) · 22 Oct 2016

Dear Referee,

Thank you very much for your comments on our joint manuscript. According to these comments, we have made corrections which hopefully could clarify the points brought up by you. We responded to these comments one by one in the attachment.

Best regards,

Jiliang Xuan, Daji Huang, Thomas Pohlmann, Jian Su, Bernhard Mayer, Ruibin Ding, Feng Zhou October 22, 2016

Response to referee

Comments: 1. The title '... in winter on the East China Sea Shelf'. In fact only results

from the winter (Feb) 2009 were analyzed and presented in the paper and no attempt was made to extend to other winters. I, therefore, suggest that the title to be changed to '. . .in the winter of 2009. . .', or similar.

Author's response: We have now analyzed the TWC structures in other winters, i.e., of the years 2010 to 2013, and added a "Figure 14" to the manuscript. Results show that the general TWC structures in the other winters were similar to that in winter 2009. Therefore, we have kept the original title "... in winter on the East China Sea Shelf".

Author's changes in manuscript: Line 896-901: A "Figure 14" was added to show the mean currents and synoptic fluctuations in the winters of the years 2010 to 2013. Line 416-421: The following discussion was added: "The simulated results in the winters of the years 2010 to 2013 (Fig. 14) show that general structures of the TWC in the other winters were similar to that in winter 2009 (Fig. 5 and Fig. 9), which indicates that the results from the winter 2009 can be regarded as representative for the winter situation. The two TWC branches and the two areas of strong fluctuations were present in all the winters from of 2009 to 2013, although their strength showed a certain inter- annual variability in accordance with the changing surface forcing and boundary fluxes.".

Comments: 2. Line 133-135, on the model, 'the river discharge of the Changjiang and Huanghe. . .'. In the immediate region of the study, there are other important rivers, e.g., Qiantang, etc.; did the authors include them; if not, why not.

Author's response: Other rivers were not included in this study, because the discharges for the other rivers are very small compared to the Changjiang and Huanghe discharge.

Author's changes in manuscript: Line 134-137: We added the statement explaining why other rivers were not included: "Other rivers were not included because of their small discharge rates, e.g., the Qiantang River, with the largest runoff from the Zhejiang coast, has a climatological mean discharge rate in winter of about 230 m3/s, which is nearly negligible compared to the Changjiang winter discharge of about 11500 m3/s.".

Comments: 3. Validation of the model. The model results were validated in the coastal regional of the East China Sea shelf and good agreements between measurements and model were obtained. However this is limited to the shallow coastal region. It is well known that the FVCOM uses a sigma co-ordinate and prone to errors in the region of steep topography. Therefore authors should at least caution readers that in the slope region, such as the Kuroshio intrusion (line 465-470), model results are less reliable unless a good validation is provided.

Author's response: We agree that our results in the region of steep topography may not have the same accuracy because of the sigma co-ordinates used in FVCOM, in particular at in the shelf break area where Kuroshio intrusion occurs. However, the synoptic fluctuations, which we focus on, are mainly located on the shelf, where results could be nicely validated. Nevertheless the validation for the offshore area could not be performed properly. Therefore, we added a statement regarding the simulation results in the slope region.

Author's changes in manuscript: Line 485-488: we added the following statement: "However, because FVCOM uses sigma co-ordinates in the vertical which are prone to errors in regions of steep topography, our results may underestimate the fluctuations at the shelf break, in particular to the northeast of Taiwan where Kuroshio intrusion occurs.".

Comments: 4. Line 264. '. . . in wintertime, both branches flowed on the isobaths, which is fully in accordance with the conservation of potential vorticity'. It implies 'cross isobaths flow is not following the pv conservation law', which is not correct. btw, 'on the isobaths' - >'along the isobaths'.

Author's response: We agree that our statement ". . . is fully in accordance with the conservation of potential vorticity" is wrong.

Author's changes in manuscript: Line 265-267: We made the following change: "In wintertime, both branches flowed on the isobaths, which is fully in accordance with the

conservation of potential vorticity" is changed to "In wintertime, both branches flowed along the isobaths, which is in accordance with the conservation of potential vorticity under frictionless conditions and for flows with a minor meridional extension".

Please also note the supplement to this comment:
http://www.ocean-sci-discuss.net/os-2016-70/os-2016-70-AC2-supplement.pdf

―――――――――――――――――――――
Interactive
comment

[Figure]

**Fig. 1.** Fig. 14

**Supplement:**

[revised manuscript text omitted]

---

## Referee Comment (RC4) · Anonymous Referee #3 · 23 Oct 2016

**General comments**

The authors aim to study the spatial distributions and dynamics of synoptic fluctuations of the Taiwan Warm Current (TWC) so as to better understand the TWC's role in winter on the cross-shelf water exchange due to the influence of its fluctuations on the regional material transport. This could be an interesting and significant scientific study. Regrettably, their analysis, interpretation and discussion are found to be incoherent and devoid of strong/convincing physical reasonings probably due to lack of comprehensive understanding of winter monsoonal flows. Some key findings of the following articles may be helpful to enhance this study:- (i) Hong, Huasheng, et al. "An overview of physical and biogeochemical processes and ecosystem dynamics in the Taiwan Strait." Continental Shelf Research 31.6 (2011): S3-S12. (ii) Hu, Jianyu, et al. "Review on current and seawater volume transport through the Taiwan Strait." Journal of Oceanography 66.5 (2010): 591-610. In addition, definitions and terms such as north of Taiwan, inshore area, inshore branch, offshore branch, alongshore, cross shore and cross shelf are noted to cause confusion when some of these terms are used interchangeably at times. Quoting literature review without further elaboration to strengthen a point is insufficient. Some figures are hard to see, not properly captioned and without the unit specified for the parameter. There exists a number of structural and grammatical errors in the language used. Finally, this is merely a case study (for January and February 2009) and the conclusions drawn are only applicable for this specific late-winter case. As such, a major revision of this manuscript, inclusive of its title, is needed before it can be considered for publication.

Major comments

1: Are November to March the winter months? How could you explain the weak mean velocity of the winter TWC on the ECS shelf (lines 51-52)? What are the dominant physical factors that cause the fluctuations of the TWC to have periods between 3 and 15 days (lines 55-58)? What do you mean by "the intermittency of the TWC in winter" (line 60)? Under what synoptic condition can the TSC be considered as an upstream flow of the TWC (line 96)? What is the physical significance of inserting "Takahashi . . . . . . the annual (?) variation of the TWC . . . .. the propagation of vorticity anomalies . . . . . ." (lines 98-100)? 2: It is obvious that your case study is for January and February 2009. Hence, your climatological (Years of climatological period are not mentioned in your manuscript) and observational deductions must refer only to these late-winter months. Apart from defining near-coast, inshore and offshore areas based on isobaths, can you offer an explanation why the TWC inshore and offshore branches only dominate in those specific isobaths (lines 75-78)? 3: Lines 103-113 under Introduction should be moved to Data and Methods (suggest to change from Methods and validation) section. 4: Please provide sufficient details on model setup, configuration, data used, forces and boundary conditions. As the model is run fully in three dimension, time steps for baroclinic and barotropic runs should be defined separately. 5: Lines

143-147: Not clear. How could you obtain the hindcast outputs for late winter 2009 when you simulated the model using 2009 to 2013 data with three years of spin-up? 6: Lines 156-171: Write-up on validations is vague. 7: Lines 186-188: "The cross-shore component (Figs 3c and 3d) is much...spatial pattern. It flows offshore in the upper layer and onshore in the lower layer at Station 1." What is seen in the figure is different from what is expressed here. 8: Errors are found in labelling Figure 4 and the explanation given on simulated and observed results is not clear. If the alongshore component is nearly one order of magnitude larger than cross-shore component, how could their fluctuation magnitudes be comparable (lines 206-207). 9: Line 235: "Second, according to the hydrostatic ..." This is ambiguous, please rewrite it. 10: Line 245: please define the mathematical form of the wind stress (at the sea surface (ta) and sea bottom (tb)) used in the model. 11: Lines 263-264: Please elaborate this statement - "...which is fully in accordance with the conservation of potential vorticity". 12: Lines 271: Explain why different cooling exists in both areas. 13: Lines 275-278: "The fact that the depth of the subsurface VMV...the effects of baroclinicity and wind friction..." Explain in detail this key finding. 14: Lines 311-312: "The currents fluctuated ...... occurred episodically". What episodic events you are referring to? 15: Line 320: Suggest you calculate the mixed layer depth based on: "Lorbacher K, Dommenget D, Niiler PP, Kohl A. Ocean mixed layer depth: A subsurface proxy of ocean atmosphere variability. J Geophys Res-Ocean. 2006; 111(C7): 1978–2012. doi: 10.1029/2003JC002157." 16: Lines 341-346: Hard to follow your explanation in the figures. Please plot them in different depths. 17: Under "3.3 Dynamic diagnostics", you argued (based on Figure 12) that the Coriolis force is mainly balanced by the total pressure in both branches,...in the wintertime TWC. This is not convincing for the Taiwan Strait. 18: Lines 395-400: I am not convinced. You argue that " ... the TSC mainly caused variations in the barotropic pressure gradients, which further ...." As I know barotropic pressure gradients is generated by a sloping sea surface and the pressure gradient is depth independent. Please clarify. 19: Lines 404-408: Confusing. Why is the negative Coriolis force associated with a northerly wind? 20: Lines 430-435: What is numerical tracer simulation and

how is it connected with tracer assimilation? Please elaborate with their physical applications. 21: Lines 491-497: Did the episodic occurrence relate to the surge and lull periods of the late winter? 22: Lines 506-519: Description and explanation are vague. 23: Lines 552-558: Confusing. Suggest to use the late winter monsoonal flow patterns during the surge and lull periods as a basis to recast your findings.

Minor Comments:

1: Line 10 : Is it 10 meter wind above sea surface ? 2: Line 30: should be "When a strong TSC intrudes towards the north of Taiwan,..." 3: Line 55: "continental shelf" instead of "continental shelves" 4: Line 120, "To investigate the currents ...ECS shelf, an unstructured-grid FVCOM was developed for..." should change to "...a 3d unstructured- grid FVCOM is developed for..." 5: Lines 124-125: "A regional refinement of the resolution (approximately 3 km) was specified ..." should be replaced by "A regional ... 3km is specified ...". 6: Line 125: Add (GEBCO) after "The General Bathymetric Chart of the Ocean". 7: Line 125 : Please specify number of grid points (m\*n\*l) used in model configuration. 8: Line 130: Usually tide is used as boundary conditions and is not a driving force. 9: Section 2.2 Validation of the mean currents and synoptic fluctuations: "The mean current was ...". What you mean by "the mean current" ? Amend "was" to " is". 10: Line 180: "...for the observational period... " Please specify the period. 11: Line 180 : "We defined the alongshore direction from southwest (218o) to northeast (38o), which is ..." It is very confusing please amend it. 12: Line 210: Section 2.3: "The Empirical ..." should be written as "The Empirical ..., as a statistical method, has been used to understand the synoptic fluctuations of the ... ". 13: Page 11, Section "2.4 Momentum analysis" should be worded as "2.4 The governing equations". 14: Page 18, Section 3.3 Dynamic diagnostics: "The seasonal mean of the water...". What is exact period of this mean. 15: Page 19, line 405: "It indicates that strong winter monsoon can weaken...". Replace "weaken" with an appropriate word. 16: Page 20, line 430: "Two section, one in the Taiwan Strait and another in ..." show transection. 17: Page 21, line 450: "Figure 14b shows the tracers

. . .". Is it tracers or traces?

Please also note the supplement to this comment:
http://www.ocean-sci-discuss.net/os-2016-70/os-2016-70-RC4-supplement.pdf

———————————————

---

## Author Comment (AC3) · 12 Nov 2016

| Ms. Ref. No.: | os-2016-70 |
| --- | --- |
| Title: | Synoptic fluctuation of the Taiwan Warm Current in winter on the East China Sea shelf |
| Journal: | Ocean Science |

Dear Referee,

Thank you very much for your comments on our joint manuscript. According to these comments, we have made corrections which hopefully could clarify the points brought up by you.

We responded to these comments one by one in the supplement file. For better examining the revisions, we combined the response with a revised manuscript in the supplement file.

Best regards,

Jiliang Xuan, Daji Huang, Thomas Pohlmann, Jian Su, Bernhard Mayer, Ruibin Ding, Feng Zhou
November 12, 2016

**Response to referee**

General comments

The authors aim to study the spatial distributions and dynamics of synoptic fluctuations of the Taiwan Warm Current (TWC) so as to better understand the TWC's role in winter on the cross-shelf water exchange due to the influence of its fluctuations on the regional material transport. This could be an interesting and significant scientific study.

**Comments: 1.** Regrettably, their analysis, interpretation and discussion are found to be incoherent and devoid of strong/convincing physical reasonings probably due to lack of comprehensive understanding of winter monsoonal flows.

**Author's response:** We have revised the manuscript according to the comments in the following two aspects: 1) unified the topic, definitions and terms throughout the manuscript, e.g., specified our topic which is about the mean currents and synoptic fluctuations in the study time (January and February 2009). 2) Explained the physical reasoning in detail, e.g., explained that other factors (wind stress and bottom stress) also play important roles on the dynamics in the Taiwan Strait.

**Author's changes in manuscript:** A number of revisions have been made according to the following major comments and minor comments.

**Comments:** 2. Some key findings of the following articles may be helpful to enhance this study:- (i) Hong, Huasheng, et al. "An overview of physical and biogeochemical processes and ecosystem dynamics in the Taiwan Strait." Continental Shelf Research 31.6 (2011): S3-S12. (ii) Hu, Jianyu, et al. "Review on current and seawater volume transport through the Taiwan Strait." Journal of Oceanography 66.5 (2010): 591-610.

**Author's response:** Thanks for the information. We have carefully studied the references and quoted them to show the dynamics of the Taiwan Strait Current.

**Author's changes in manuscript:** in **line 105-107**, we added a statement: "Hong et al. (2011) and Hu et al. (2010) summarized that the temporal and spatial variation of TSC is modulated by strong monsoon forcing, complex topography and circulation in the northern South China Sea as well as coastal water input and the Kuroshhio intrusion".

**Comments:** 3. In addition, definitions and terms such as north of Taiwan, inshore area, inshore branch, offshore branch, alongshore, cross shore and cross shelf are noted to cause confusion when some of these terms are used interchangeably at times.

**Author's response:** We have now unified the terms in two aspects: 1) Since the study area is on the ECS shelf, we changed "cross-shelf" to "cross-shore" throughout the manuscript; 2) explained that the inshore branch is associated with the inshore area.

**Author's changes in manuscript:** We changed "cross-shelf" to "cross-shore" throughout the manuscript. Moreover, in **line 82-83**, we defined that "the inshore area is the area between the 30 and 100 m isobaths where the TWC inshore branch

dominates".

**Comments:** 4. Quoting literature review without further elaboration to strengthen a point is insufficient. Some figures are hard to see, not properly captioned and without the unit specified for the parameter. There exists a number of structural and grammatical errors in the language used.
**Author's response:** We have revised the manuscript according to the comments.
**Author's changes in manuscript:** A number of revisions have been made according to the following major comments and minor comments.

**Comments:** 5. Finally, this is merely a case study (for January and February 2009) and the conclusions drawn are only applicable for this specific late-winter case. As such, a major revision of this manuscript, inclusive of its title, is needed before it can be considered for publication.
**Author's response:** We have now analyzed the TWC structures in the following four winters as well, i.e., December-March of the years 2010 to 2013, and added a "Figure 14" to the manuscript. Results show that the general TWC structures in the other winters were similar to that in January-February of 2009. Therefore, the results from the January-February 2009 can be regarded as representative for the winter situation.
**Author's changes in manuscript: Line 943-947:** A "Figure 14" was added to show the mean currents and synoptic fluctuations in the winters of the years 2010 to 2013.
**Line 443-450:** The following discussion was added: "The simulated results in the winters of the years 2010 to 2013 (Fig. 14) show that general structures of the TWC in the other winters were similar to that in winter 2009 (Fig. 5 and Fig. 9), which indicates that the results from the winter 2009 can be regarded as representative for the winter situation. The two TWC branches and the two areas of strong fluctuations were present in all the winters from of 2009 to 2013, although their strength showed a certain inter- annual variability in accordance with the changing surface forcing and boundary fluxes".

Major comments
**Comments:** 1-1. Are November to March the winter months?
**Author's response:** We have now added a definition of winter months from December to March, according to the critical value of local air temperature (< 10 ℃ in winter) in the East China Sea (Chen et al. 1992). Detail information could be seen in the reference: Shangji Chen, Weihuan He, Tiyu Yao, et al., 1992, Discrimination of ocean hydroclimate seasons in the China Seas, Acta Oceanologica Sinica, 14(6), 1-11 (In Chinese).
**Author's changes in manuscript: Line 50:** We defined the wintertime as December to March.

**Comments:** 1-2. How could you explain the weak mean velocity of the winter TWC on the ECS shelf (lines 51-52)?

**Author's response:** We agree that the statement of "weak mean velocity" is not clear. We have changed "weak mean velocity" to "weak mean surface velocity" according to Qiu and Imasato (1990)'s study. Qiu and Imasato (1990)'s study is also quoted to show the climatological structure of the surface current in the East China Sea, which is mapped by averaging GEK current data available over 1/5 ° × 1/5 ° resolution boxes from 1953 to 1984.

**Author's changes in manuscript: Line 51-52:** We changed the statement "… its weak mean velocity superimposed by pronounced small-scale spatial and synoptic temporal variations" to "… its weak mean surface velocity, according to a climatological structure of the surface current in the ECS mapped by Qiu and Imasato (1990)".

**Comments:** 1-3. What are the dominant physical factors that cause the fluctuations of the TWC to have periods between 3 and 15 days (lines 55-58)?

**Author's response:** The physical mechanism of the TWC fluctuations is a question to be solved in our manuscript. In the former studies, the wind was recognized as a main physical factor over most continental shelves. In the coastal area of the ECS, Huang et al. (2016) have shown that the wind was a main physical factor which caused the temporal variation of the wintertime currents at the synoptic scale. However, as the Hong et al. (2011)'s study (mentioned by the referee) said that "Due to its complex bottom topography, alternating monsoon forcing and conjunction of several current systems [such as the Zhejiang–Fujian (Zhe–Min) Coastal Current, the Kuroshio intrusion and the extension of the South China Sea Warm Current], the physical and biogeochemical processes and ecosystem dynamics in the Taiwan Strait vary significantly both in space and in time", the dominant physical factors which cause the TWC fluctuations on the whole shelf of the ECS may be very complicated. Therefore, we added an argument to raise the question that the dominant physical factors of the TWC fluctuations are still lack of study.

**Author's changes in manuscript: Line 58-63:** The following argument was added: "Huang et al. (2016) have shown that the wind was a main physical factor which caused the temporal variation of the wintertime currents at the synoptic scale in the coastal area of the ECS. However, the dominant physical factors of the TWC fluctuations are still lack of study, regarding that the fluctuations on the whole shelf of the ECS may be more complicated due to the complex bottom topography, alternating monsoon forcing and conjunction of several current systems such as the Kuroshio Current, the Taiwan Strait Current (TSC) and the Zhe-Min Coastal Current (ZMCC)."

**Comments:** 1-4. What do you mean by "the intermittency of the TWC in winter" (line 60)?

**Author's response:** We agree that the statement of "the intermittency of the TWC in winter" is not clear. Zhu et al. (2004)'s study demonstrated that the TWC has an episodic wintertime feature. This means the TWC intermittently occurs in winter. Therefore, we referred Zhu et al. (2004)'s result to illustrate the episodic wintertime feature of the TWC.

**Author's changes in manuscript: Line 65-67:** We changed "some recent observations have shown that the intermittency of the TWC in winter reaches maximum velocity variations larger than 0.2 m/s (Zhu et al., 2004; Zeng et al., 2012)" to "some recent observations have shown that the TWC has an episodic wintertime feature (Zhu et al., 2004) and the intermittency of the TWC in winter has an amplitude as large as 0.2 m/s (Zeng et al., 2012)".

**Comments:** 1-5. Under what synoptic condition can the TSC be considered as an upstream flow of the TWC (line 96)?
**Author's response:** Obviously it is the northeastward TSC that could be an upstream flow of the TWC. In addition, we have referred Hong et al. (2011)'s overview to explain the physical factors of the TSC variations.
**Author's changes in manuscript: Line 104:** Changed "The TSC" to "The northeastward TSC"; **Line 105-107:** we added a result of Hong et al. (2011)'s overviews "Hong et al. (2011) and Hu et al. (2010) summarized that the temporal and spatial variation of TSC is modulated by strong monsoon forcing, complex topography and circulation in the northern South China Sea as well as coastal water input and the Kuroshhio intrusion.".

**Comments:** 1-6. What is the physical significance of inserting "Takahashi … the annual (?) variation of the TWC … the propagation of vorticity anomalies …" (lines 98-100)?
**Author's response:** We have now added a statement to explain the significance of Takahhashi and Morimoto (2013)'s result, which further demonstrated that the fluctuation of TWC was associated with its upstream currents such as the TSC.
**Author's changes in manuscript: Line 109-112:** we revised the statement as following: "Takahashi and Morimoto (2013) pointed out that the temporal variation of the TWC is characterized by the propagation of vorticity anomalies originating from northeast of the Taiwan Strait, which further demonstrated that the fluctuation of TWC was associated with its upstream currents such as the TSC".

**Comments:** 2-1. It is obvious that your case study is for January and February 2009. Hence, your climatological (Years of climatological period are not mentioned in your manuscript) and observational deductions must refer only to these late-winter months.
**Author's response:** We have now analyzed the TWC structures in the following four winters as well, i.e., December-March of the years 2010 to 2013, and added a "Figure 14" to the manuscript. Results show that the general TWC structures in the other winters were similar to that in January-February of 2009. Therefore, the results from the January-February 2009 can be regarded as representative for the winter situation.
**Author's changes in manuscript: Line 943-947:** A "Figure 14" was added to show the mean currents and synoptic fluctuations in the winters of the years 2010 to 2013.
**Line 443-450:** The following discussion was added: "The simulated results in the winters of the years 2010 to 2013 (Fig. 14) show that general structures of the TWC in the other winters were similar to that in winter 2009 (Fig. 5 and Fig. 9), which

indicates that the results from the winter 2009 can be regarded as representative for the winter situation. The two TWC branches and the two areas of strong fluctuations were presented in the all winters from of 2009 to 2013, although their strength showed a certain inter- annual variability in accordance with the changing surface forcing and boundary fluxes".

**Comments:** 2-2. Apart from defining near-coast, inshore and offshore areas based on isobaths, can you offer an explanation why the TWC inshore and offshore branches only dominate in those specific isobaths (lines 75-78)?

**Author's response:** We have now added an explanation that the TWC inshore and offshore branches mainly occur in those specific isobaths is due to the conservation of potential vorticity, according to the hydrographic data analysis and numerical interpretation by Su and Pan (1987).

**Author's changes in manuscript: Line 84-86:** We added the explanation that "According to the hydrographic data analysis and numerical interpretation by Su and Pan (1987), the TWC inshore and offshore branches mainly occur in those specific isobaths is due to the conservation of potential vorticity".

**Comments:** 3-1. Lines 103-113 under Introduction should be moved to Data and Methods (suggest to change from Methods and validation) section.

**Author's response:** We still kept this paragraph under the section of Introduction for the following reasons: 1) statements in the paragraph are an introduction of three popular methods in studying the synoptic fluctuations of the wintertime TWC, while the first two methods were not used in this manuscript because the observation data are limited in terms of temporal and spatial coverage; 2) statements in the paragraph indicated that the numerical simulation provide a promising approach for studying the synoptic fluctuations, which gave the background for the next section of Methods and validation. In addition, the section name "Methods and validation" was also kept because the model validation is a very important part in the manuscript since we need reliable numerical simulations to investigate the TWC fluctuations.

**Author's changes in manuscript:** Nothing has been changed, due to the above arguments.

**Comments:** 4-1. Please provide sufficient details on model setup, configuration, data used, forces and boundary conditions. As the model is run fully in three dimension, time steps for baroclinic and barotropic runs should be defined separately.

**Author's response:** We have carefully examined the model setup, configuration and driving forces and added some essential information.

**Author's changes in manuscript: Line 134:** We changed "an unstructured-grid FVCOM" to "an unstructured-grid (Fig. 2, left panel) FVCOM". **Line 145:** We changed "The river discharge" to "The daily-mean river discharge". **Line 147-150:** We added a statement "Other rivers were not included because of their small discharges, e.g., the Qiantang River, with the largest runoff from the Zhejiang coast, has a climatological mean discharge in winter of about 230 $m^3$/s, which is nearly

negligible compared to the Changjiang winter discharge of about 11500 m$^3$/s". **Line 161-162:** We changed "The model time step was 90 seconds" to "The model time step was 15 seconds for the 2-D barotropic mode and 90 seconds for the 3-D baroclinic mode".

**Comments:** 5-1. Lines 143-147: Not clear. How could you obtain the hindcast outputs for late winter 2009 when you simulated the model using 2009 to 2013 data with three years of spin-up?

**Author's response:** We have now added the detailed information for the spin-up years (2006-2008) and the initial conditions (the initial temperature and salinity were taken from the Hybrid Coordinate Ocean Model and initial velocity was set to zero).

**Author's changes in manuscript: Line 159-160:** We added the detailed information for the spin-up years and the initial conditions as following: "… simulation from 2009 to 2013 are used, following three spin-up years (2006-2008) initiated with the temperature and salinity taken from the Hybrid Coordinate Ocean Model and velocity setted to zero".

**Comments:** 6-1. Lines 156-171: Write-up on validations is vague.

**Author's response:** We have rewritten the validations of circulation structure and boundary fluxes.

**Author's changes in manuscript: Line 173:** We emphasized that "The FVCOM has reproduced almost all of the known circulation structure in the ECS in winter". **Line 182-184:** We highlighted the validation of the volume transport through the Taiwan Strait that "The simulated transports were accurate enough to reproduce volume transport (1.22 Sv) through the Taiwan Strait which is closer to the observation value (1.20 Sv) from Isobe (2008) than former model results".

**Comments:** 7-1. Lines 186-188: "The cross-shore component (Figs 3c and 3d) is much … spatial pattern. It flows offshore in the upper layer and onshore in the lower layer at Station 1." What is seen in the figure is different from what is expressed here.

**Author's response:** It is because the color scales in Fig. 3 are not same between the alongshore components and cross-shore components.

**Author's changes in manuscript: Line 875-880:** We added a notation that "Note, an enlarged color scale is used for the cross-shore component to have a clear view of its weak structure.".

**Comments:** 8-1. Errors are found in labelling Figure 4 and the explanation given on simulated and observed results is not clear.

**Author's response:** We have now revised the label and explanation of Figure 4.

**Author's changes in manuscript: Line 882:** We changed "Validations of the wintertime TWC fluctuations" to "Variations of the inshore branch of TWC during January and February 2009". **Line 218-219:** We changed "Synoptic fluctuations of the wintertime TWC were also validated against the mooring results (Fig. 4)" to "Synoptic fluctuations of the TWC inshore branch during January and February 2009

were also validated against the mooring results (Fig. 4)".

**Comments:** 8-2. If the alongshore component is nearly one order of magnitude larger than cross-shore component, how could their fluctuation magnitudes be comparable (lines 206-207).
**Author's response:** Apparently "in the mean condition" was missing in the first sentence "the alongshore component is nearly one order of magnitude larger than cross-shore component".
**Author's changes in manuscript: Line 227-228:** We revised the statement as following: "…the alongshore component is nearly one order of magnitude larger than the cross-shore component in the mean condition …"

**Comments:** 9-1. Line 235: "Second, according to the hydrostatic …" This is ambiguous, please rewrite it.
**Author's response:** We have rewritten the sentence.
**Author's changes in manuscript: Line 256-257:** We changed "Second, according to the hydrostatic assumption used in FVCOM [as shown in Eq. (2)], the pressure is integrated from depth z to the sea surface" to "Second, according to the hydrostatic approximation used in FVCOM [as shown in Eq. (2)], the pressure gradient is given as the product of density times the gravitational acceleration".

**Comments:** 10-1. Line 245: please define the mathematical form of the wind stress (at the sea surface (ta) and sea bottom (tb)) used in the model.
**Author's response:** We have now defined the mathematical form of the wind stress and bottom stress.
**Author's changes in manuscript: Line 265-270:** Eq. (5) has been revised to show the mathematical form of the wind stress $\underbrace{\rho_a C_D \left|\vec{U}\right|\vec{U}}_{\tau_a}$ and bottom stress $\underbrace{-k_b \left|\vec{U_b}\right|\vec{U_b}}_{\tau_b}$.

**Comments:** 11-1. Lines 263-264: Please elaborate this statement - "…which is fully in accordance with the conservation of potential vorticity".
**Author's response:** We agree that our statement "… is fully in accordance with the conservation of potential vorticity" is not correct due to the influence of frictions.
**Author's changes in manuscript: Line 288-290:** We made the following change: "In wintertime, both branches flowed on the isobaths, which is fully in accordance with the conservation of potential vorticity" is changed to "In wintertime, both branches flowed along the isobaths, which is in accordance with the conservation of potential vorticity under frictionless conditions and for flows with a minor meridional extension".

**Comments:** 12-1. Lines 271: Explain why different cooling exists in both areas.
**Author's response:** The differential cooling between coastal and offshore area is due to the different heat capacity of water columns with different water depths.

**Author's changes in manuscript: Line 297-300:** We explained that "Assuming a relatively spatially homogeneous heat loss, a different cooling occurs, due to the smaller heat capacity of the shallow coastal water compared to the deep offshore water".

**Comments:** 13-1. Lines 275-278: "The fact that the depth of the subsurface VMV … the effects of baroclinicity and wind friction …" Explain in detail this key finding.

**Author's response:** We have explained this point (the effects of baroclinicity and wind friction on the inshore branch are stronger than for the offshore branch) in **line 296-300:** 1) The northerly wind in winter weakens the northward TWC, particularly in the upper layer, which leads to the formation of the subsurface VMV; 2) Differential cooling, due to different heat capacity of the water columns, as explained in the previous response is responsible for a stronger cooling of the coastal shallow waters ccompared to the offshore deep waters. In addition, through analyzing the vertical structure of the inshore branch (P2, Fig. 7) in the following section (3.2 Synoptic fluctuations, Line 343-353), we also reached the conclusion that the fluctuations induced by wind and surface cooling are stronger in the inshore branch than that in the offshore branch.

**Author's changes in manuscript:** We hope that the clarification with respect to the previous comment (12-1) helps to make our point more clear. Moreover, we have modified the text at lines 302 to 304 and tried put a stronger focus on our main findings.

**Comments:** 14-1. Lines 311-312: "The currents fluctuated … occurred episodically". What episodic events you are referring to?

**Author's response:** From the result of the standard deviations of the currents (Fig. 7), we can infer that the TWC inshore branch occurs episodically, although specific episodic events could not be identified. However, our manuscript is intended to focus on the synoptic variations (with about 3-15 days periods) and their general impact on the water transports, which is different from the traditional concerns only on mean structures. Hence, specific events have not been highlighted in the manuscript except, for two extreme events in the discussion.

**Author's changes in manuscript: Line 338-339:** We changed "The currents fluctuated in the alongshore direction …" to "As deduced from the standard deviation, the currents fluctuated significantly in the alongshore direction …"

**Comments:** 15-1. Line 320: Suggest you calculate the mixed layer depth based on: "Lorbacher K, Dommenget D, Niiler PP, Kohl A. Ocean mixed layer depth: A subsurface proxy of ocean atmosphere variability. J Geophys Res-Ocean. 2006; 111(C7): 1978–2012. doi: 10.1029/2003JC002157."

**Author's response:** Thank you for your information about how to calculate the upper mixed layer depth from observed temperature, salinity or potential density. In this manuscript, we kept the original method, which used the critical Richardson number, because it is based on the dynamics of instability which include both effect of the

potential density and vertical current shear and is also implemented in the FVCOM. This method is also widely used, see e.g. Mellor and Durbin (1975), Grachev et al., 2013, and Richardson et al., (2013).

Mellor, G. L. and Durbin, P A.: The structure and dynamics of the ocean surface mixed layer. J. Phys. Oceanogr., 5(4), 718–728, 1975.

Grachev, A. A., Andreas, E. L., Fairall, C. W., Guest, P. S., and Persson, P. O. G.: The critical Richardson number and limits of applicability of local similarity theory in the stable boundary layer, Boundary-layer meteorology, 147(1), 51–82, 2013.

Richardson, H., Basu, S., and Holtslag, A. A. M.: Improving stable boundary-layer height estimation using a stability-dependent critical bulk Richardson number, Boundary-layer meteorology, 148(1), 93–109, 2013.

**Author's changes in manuscript: Line 346-348:** We have quoted the references as the following: "The depths of the upper mixed layer were determined by a Richardson number criterion (Mellor and Durbin, 1975; Grachev et al., 2013; Richardson et al., 2013), where the critical Richardson number equals 0.25 in this paper [as in Xuan et al. (2012)]"

**Comments:** 16-1. Lines 341-346: Hard to follow your explanation in the figures. Please plot them in different depths.

**Author's response:** We have now added the 30, 50, 70, 100 and 200 m isobaths in Fig. 9.

**Author's changes in manuscript: Line 914-917:** We redrawn the Fig. 9 by adding the 30, 50, 70, 100 and 200 m isobaths.

**Comments:** 17-1. Under "3.3 Dynamic diagnostics", you argued (based on Figure 12) that the Coriolis force is mainly balanced by the total pressure in both branches, … in the wintertime TWC. This is not convincing for the Taiwan Strait.

**Author's response:** We agree that other factors also play important roles on the dynamics in the Taiwan Strait, especially the wind stress and bottom stress. Both the studies of Guo et al. (2003, Fig. 13) and this manuscript (Fig. 12) showed the important effects of wind stress and bottom stress in the Taiwan Strait.

**Author's changes in manuscript: Line 404-407:** We have added the effects of wind stress and bottom stress in the Taiwan Strait as follows: "the wind-induced surface friction plays an important role in the TWC, especially in the inshore area and the Taiwan Strait (Fig. 12c). The bottom friction has an impact north of Taiwan and in the shallow Taiwan Strait, in particular when a significant Kuroshio intrusion enhances the bottom flow (Fig. 12d)."

**Comments:** 18-1. Lines 395-400: I am not convinced. You argue that "… the TSC mainly caused variations in the barotropic pressure gradients, which further …" As I know barotropic pressure gradients is generated by a sloping sea surface and the pressure gradient is depth independent. Please clarify.

**Author's response:** We agree that barotropic pressure gradients are generated by a sloping sea surface and the pressure gradient is depth independent. In **line 420-424**,

we have explained, that "The mechanism can be interpreted as follows. When a larger TSC intrusion occurred, the isobaric slope tilted downward from south to north, generating a cross-shore current from the coastal area to the offshore area. On the contrary, when the TSC intrusion was weak, the Kuroshio intrusion from offshore to inshore dominated north of Taiwan".

**Author's changes in manuscript: Line 425-426:** We changed "the TSC mainly caused variations in the barotropic pressure gradients" to "in the shallow coastal area the TSC mainly caused variations in the depth-independent barotropic pressure gradients".

**Comments:** 19-1. Lines 404-408: Confusing. Why is the negative Coriolis force associated with a northerly wind?

**Author's response:** We agree that the statement of negative Coriolis force is not clear. Obviously it is the northwestward Coriolis force that indicates the southwestward current.

**Author's changes in manuscript: Line 435:** We changed "negative" to "northwestward direction".

**Comments:** 20-1. Lines 430-435: What is numerical tracer simulation and how is it connected with tracer assimilation? Please elaborate with their physical applications.

**Author's response:** We have now added a detailed statement for the numerical tracer simulation and its associated physical applications.

**Author's changes in manuscript: Line 469-474**: We modified the text as follows: "A numerical tracer simulation was used to analyze the role of the cross-shore fluctuation in the transport of the TSC water and the Kuroshio water north of Taiwan. In order to demonstrate the characteristics of the flow patterns more clearly, artificial tracers are released in the model domain and transported by the velocity field provided by the FVCOM simulation. The tracer running was part of the FVCOM simulation; therefore, all the above mentioned dynamics were involved, e.g., tide, wind, and boundary forces."

**Comments:** 21-2. Lines 491-497: Did the episodic occurrence relate to the surge and lull periods of the late winter?

**Author's response:** No, the episodic occurrence of the TWC inshore branch was related to the wind-induced synoptic fluctuations rather than the wind surge. The difference between the wind-induced synoptic fluctuations and wind surge can be explained by two factors: 1) The synoptic fluctuations in the inshore branch are induced by the variation of wind speed and direction, while the wind/storm surge is caused by large wind speed regardless of wind direction. 2) The wind-induced synoptic fluctuations has a period of 3-15 days while the storm surge occurs occasionally without a specific period, hence, the wind-induced synoptic fluctuations occur more frequently than storm surge events. Therefore, we study the episodic occurrence induced by the synoptic fluctuations rather than extreme surges and lull snapshots.

**Author's changes in manuscript: Line 539-541:**   we have added a statement that "In this paper, only wind-induced synoptic fluctuations are considered in the relations to the episodic events and no short-term extreme storm events."

**Comments:** 22-3. Lines 506-519: Description and explanation are vague.
**Author's response:** We have now added some information on the description and explanation in the discussion of the offshore transports along the latitudes 26.5 °N and 28 °N.
**Author's changes in manuscript: Line 554:** We have added a quote of Fig. 9f to specify the locations of the offshore transports. **Line 560-561:** We changed the statement "… played an important role in cross-shelf material exchange. Several mechanisms have been used to explain this process" to "… played an important role in cross-shelf material exchange, but the mechanisms of the offshore-penetrating fronts are still under debate".

**Comments:** 23-1. Lines 552-558: Confusing. Suggest to use the late winter monsoonal flow patterns during the surge and lull periods as a basis to recast your findings.
**Author's response:** we kept the statement of synoptic fluctuations because it is different from the surge and lull periods. Here we concluded the roles of synoptic fluctuations on the alongshore and cross-shore transports in two areas, north of Taiwan and in the inshore area, respectively.
**Author's changes in manuscript: Line 601:** See our response to comment 21-2. Moreover, we changed "Due to these fluctuations" to "Due to the fluctuation north of Taiwan".

Minor Comments:
**Comments:** 1: Line 10 : Is it 10 meter wind above sea surface ?
**Author's response:** Yes, it is.
**Author's changes in manuscript: Line 151:** We have shown the source of wind data as following: "the 3-hourly wind stress and 10 m wind speed data was from the ERA-40 re-analysis (Uppala et al., 2005)".

**Comments:** 2: Line 30: should be "When a strong TSC intrudes towards the north of Taiwan, …"
**Author's response:** Thanks.
**Author's changes in manuscript: Line 30:** We changed "When a larger TSC intrudes north of Taiwan" to "When a strong TSC intrudes to the north of Taiwan".

**Comments:** 3: Line 55: "continental shelf" instead of "continental shelves"
**Author's response:** Since the "continental shelves" is related to the "some features common", we kept the plural "shelves".
**Author's changes in manuscript:** Nothing has been changed.

**Comments:** 4: Line 120, "To investigate the currents … ECS shelf, an unstructured-grid FVCOM was developed for …" should change to "… a 3d unstructured- grid FVCOM is developed for …"

**Author's response:** Agree.

**Author's changes in manuscript: Line 134:** We changed "an unstructured-grid … was developed" to "a 3-D unstructured-grid … is developed".

**Comments:** 5: Lines 124-125: "A regional refinement of the resolution (approximately 3 km) was specified …" should be replaced by "A regional … 3km is specified …".

**Author's response:** Agree.

**Author's changes in manuscript: Line 136:** We changed "was" to "is".

**Comments:** 6: Line 125: Add (GEBCO) after "The General Bathymetric Chart of the Ocean".

**Author's response:** Agree.

**Author's changes in manuscript: Line 138:** We added "(GEBCO)".

**Comments:** 7: Line 125: Please specify number of grid points (m*n*l) used in model configuration.

**Author's response:** We have now added the cell number (n), which is the most important index in estimating the FVCOM quality.

**Author's changes in manuscript: Line 139:** We changed "Twenty vertical layers" to "Twenty vertical layers with 76954 triangle cells".

**Comments:** 8: Line 130: Usually tide is used as boundary conditions and is not a driving force.

**Author's response:** We kept the definition of tides as driving forces, because according to our understanding external forces can been imposed not only at the air-sea boundary (wind stress, heat fluxes), but also at lateral boundaries.

**Author's changes in manuscript:** Nothing has been changed.

**Comments:** 9: Section 2.2 Validation of the mean currents and synoptic fluctuations: "The mean current was …". What you mean by "the mean current" ? Amend "was" to " is".

**Author's response:** We have now specified the mean currents as the Kuroshio Current, the TWC, and the ZMCC.

**Author's changes in manuscript: Line169:** Revised as following: "The mean currents, e.g., the Kuroshio Current, the TWC, and the ZMCC".

**Comments:** 10: Line 180: "…for the observational period … " Please specify the period.

**Author's response:** We have specified the period: "… for the observational period,

which was from January 1 to February 28, 2009".

**Author's changes in manuscript:** We changed "…for the observational period … " to "…for the observational period , which was from January 1 to February 28, 2009"

**Comments:** 11: Line 180: "We defined the alongshore direction from southwest (218o) to northeast (38o), which is …" It is very confusing please amend it.

**Author's response:** We have now revised the statement.

**Author's changes in manuscript: Line 201-202:** We changed the statement as follows: "Using the same method as in Huang et al. (2016), we defined the positive alongshore current directing from southwest (218 ֯) to northeast (38 ֯)".

**Comments:** 12: Line 210: Section 2.3: "The Empirical …" should be written as "The Empirical …, as a statistical method, has been used to understand the synoptic fluctuations of the …".

**Author's response:** Agree.

**Author's changes in manuscript: Line 234:** We changed "The Empirical Orthogonal Function (EOF) method (Emery and Thomson, 2001)" to "The Empirical Orthogonal Function (EOF) method (Emery and Thomson, 2001), as a statistical method".

**Comments:** 13: Page 11, Section "2.4 Momentum analysis" should be worded as "2.4 The governing equations".

**Author's response:** We kept the statement of "2.4 Momentum analysis". As a method name, the "2.4 Momentum analysis" is more appropriate than the "2.4 The governing equations". In addition, it is consistent with the name of upper section "2.3 EOF analysis of synoptic fluctuations".

**Author's changes in manuscript:** Nothing has been changed.

**Comments:** 14: Page 18, Section 3.3 Dynamic diagnostics: "The seasonal mean of the water …". What is exact period of this mean.

**Author's response:** Thanks.

**Author's changes in manuscript: Line 401:** We changed "The seasonal mean" to "The wintertime (January and February 2009) mean".

**Comments:** 15: Page 19, line 405: "It indicates that strong winter monsoon can weaken …". Replace "weaken" with an appropriate word.

**Author's response:** We have changed the statement of this sentence.

**Author's changes in manuscript: Line 437:** We changed "strong winter monsoon can weaken or even stop the TWC" to "strong northerly monsoon in winter can reduce or even stop the northeastward TWC".

**Comments:** 16: Page 20, line 430: "Two section, one in the Taiwan Strait and another in …" show transection.

**Author's response:** We have added the quote of Fig. 15 which showed the two

sections.

**Author's changes in manuscript: Line 474-475:** We changed the statement as follows: "Two sections, one in the Taiwan Strait (Fig. 15a, black dots) and another in the East Taiwan Channel (Fig. 15b, black dots)".

**Comments:** 17: Page 21, line 450: "Figure 14b shows the tracers …". Is it tracers or traces?

**Author's response:** Thanks. It is traces.

**Author's changes in manuscript: Line 491:** We changed "tracers" to "traces".

[revised manuscript text omitted]

---

## Author Response (AR2)

| Ms. Ref. No.: | os-2016-70 |
|---|---|
| Title: | Synoptic fluctuation of the Taiwan Warm Current in winter on the East China Sea shelf |
| Journal: | Ocean Science |

Dear Dr. John M. Huthnance,

Thanks very much for your comments and assistances in editing our joint manuscript. We have made all the related corrections on the manuscript according to your comments.

Appended to this letter is the "Response to the referee" and a "marked-up manuscript". The "Response to the referee" contains our point-by-point responses to the comments raised by you. The "marked-up manuscript" tracks our changes.

Once again, thank you very much for your kind support which significantly improved our manuscript. On behalf of all authors, we would like to express our great appreciation to you and the reviewers by mentioning all of you in the Acknowledgements.

Yours sincerely,

Jiliang Xuan, Daji Huang, Thomas Pohlmann, Jian Su, Bernhard Mayer, Ruibin Ding, Feng Zhou
January 1, 2017

**Response to the referee**

**Firstly a question. You are discussing winter but refer to the south-west "monsoon". Is "monsoon" the correct word in winter as well as summer, or should it be "wind"? [North-east monsoon is winter is OK, I think].**

**Author's response:** We agree that the statement of "southwesterly monsoon in winter" is wrong in the manuscript. Revisions have been made in two aspects:

1) The word "monsoon" was changed to "wind" when discussing the southwesterly wind in winter.

2) The statements relating to wind direction and current direction throughout the manuscript were modified to be more accurately according to their actual directions, e.g., "northerly monsoon" was changed to "northeasterly monsoon", "northward TWC" was changed to "northeastward TWC" etc.

**Author's changes in manuscript:** We have revised the above mentioned "words" and "phrases" throughout the manuscript.

(Abstract)

**Line 34. Omit ", extending in the region"**

**Author's response:** Agree.

**Author's changes in manuscript:** Line 33: Omit ", extending in the region".

**Lines 34-35. "fluctuations . . alongshore . . are important for . . cross-shore transports." This reads strangely.**

**Author's response:** We have now revised the statement.

**Author's changes in manuscript:** Line 34-35: "The fluctuations are generally strong in the alongshore direction, in particular at the latitudes 26.5 N and 28 N where they are important for the local cross-shore transports" **was changed to** "The fluctuations are generally strong both in the alongshore and cross-shore directions, in particular at the latitudes 26.5 N and 28 N".

(1. Introduction)

**Line 51. Better ". . its weak mean surface velocity . ."**

**Author's response:** Agree.

**Author's changes in manuscript:** Line 51: "its surface weak mean velocity" was changed to "its weak mean surface velocity".

**Lines 60-61. Better ". . fluctuations still lack study; the fluctuations . . may be complicated"**

**Author's response:** Agree.

**Author's changes in manuscript:** Line 61-62: "fluctuations are still lack of study, regarding that the fluctuations on the whole shelf of the ECS may be more complicated due to the complex bottom topography" **was changed to** "fluctuations still lack study; the fluctuations on the whole shelf of the ECS may be complicated due to the complex bottom topography".

**Line 67. Intermittency does not have an amplitude: ". . the variations of the TWC . . have an amplitude . ."**
**Author's response:** Thanks.
**Author's changes in manuscript:** Line 67: "intermittency" was changed to "variations", and "has" was changed to "have".

**Line 68. You might want to change this to: ". . the variations of the TWC . . cause a . ."**
**Author's response:** Agree.
**Author's changes in manuscript:** Line 68: "intermittency" was changed to "variations", and "causes" was changed to "cause".

**Line 86. "mainly occur close to those specific isobaths. . ."**
**Author's response:** Agree.
**Author's changes in manuscript:** Line 86: "mainly occur in those specific isobaths" was changed to "mainly occur close to those specific isobaths".

**Lines 93-94. ". .indicating a strong synoptic fluctuation . ." ["stronger" implies a comparison which you do not make]**
**Author's response:** Thanks.
**Author's changes in manuscript:** Line 93-94: "much stronger" was changed to "strong".

**Lines 124-125. Better ". . investigate wintertime TWC synoptic fluctuations and their mechanisms. The rest . ."**
**Author's response:** Agree.
**Author's changes in manuscript:** Line 125: "synoptic fluctuations and their mechanisms of the wintertime TWC" was changed to "wintertime TWC synoptic fluctuations and their mechanisms".

(2.1)
**Line 160. ". . velocity set to zero. . ."**
**Author's response:** Agree.
**Author's changes in manuscript:** Line 160: "setted" was changed to "set".

**Line 161. ". . and at the lateral boundaries a sponge layer"**
**Author's response:** Agree.
**Author's changes in manuscript:** Line 161: "and that at the lateral boundaries sponge layer was used" was changed to "and at the lateral boundaries a sponge layer was used".

**Line 163. Delete "of"**
**Author's response:** Agree.
**Author's changes in manuscript:** Line 163: "of" was deleted.

(2.2)
**Line 204. "direction as from . ."**
**Author's response:** Agree.
**Author's changes in manuscript:** Line 204: "directing" was to changed "direction as".

**Lines 205-206. ". . cross-shore direction is from northwest . .(128 °), normal to the isobaths. The alongshore . ."**
**Author's response:** Agree.
**Author's changes in manuscript:** Line 205-206: "The positive cross-shore direction is the mean normal direction of the isobaths from northwest (308 °) to southeast (128 °)" **was changed to** "The positive cross-shore direction is from northwest (308 °) to southeast (128 °), normal to the isbaths".

**Lines 229-230. ". . i.e. that the mean alongshore component . . than the mean cross-shore component, the magnitude . ." Maybe the last part of this sentence "the magnitude . . alongshore fluctuations" should move to the beginning of the sentence.**
**Author's response:** We have now revised the statement.
**Author's changes in manuscript:** Line 228-232: "In contrast to the anisotropic feature for the mean currents (Fig. 3), i.e., that the alongshore component is nearly one order of magnitude larger than the cross-shore component in the mean condition, the magnitude of the cross-shore fluctuations is comparable to the alongshore fluctuations" **was changed to** "The magnitude of the cross-shore fluctuations is comparable to the alongshore fluctuations. This is different to the anisotropic characteristic of the mean currents (Fig. 3), for which the alongshore component is nearly one order of magnitude larger than the cross-shore component".

(2.4)
**Line 269. "comparably" -> "comparatively"**
**Author's response:** Agree.
**Author's changes in manuscript:** Line 270: "comparably" was changed to "comparatively".

**Equation (5) and line 273. Velocity at the bottom might better be vector "Vb".**
**Author's response:** Agree.
**Author's changes in manuscript:** Line 271-274: "$U_b$" was changed to "$V_b$".

(3.1).
**Line 285. Can omit "a horizontal structure with"**

**Author's response:** Agree.

**Author's changes in manuscript:** Line 285: "a horizontal structure with" was deleted.

**Line 297. Delete final "a"**

**Author's response:** Agree.

**Author's changes in manuscript:** Line 298: "a different cooling occurs" was changed to "different cooling occurs".

**Lines 298 to 301. I am confused by the directions here. If "offshore" (line 299) is to the south-east then the vertical shear would be northeastward (not as stated on line 299), which does result in a southwestward flow component increasing downwards. Whether this weakens the northeastward TWC depends on assumptions about where in depth the flow is unchanged. This also needs better explanation to justify "effects of baroclinicity" in line 303.**

**Author's response:** Agree with your argument. A northeastward thermal current (vertical shear of current) is generated by a northwestward density gradient, which results in a northeastward flow (TWC) increasing upwards.

The subsurface current core is caused by the combined effects of wind friction (which results in a southwestward flow increasing downward) and baroclinicity (which results in a northeastward flow increasing upward).

**Author's changes in manuscript:**

**Line 292-308:** The whole paragraph was rewritten as follows. "We further examined the subsurface current core using the depth of the VMV (Fig. 5b). We found that the VMV of the TWC was located 40–60 m below the surface at the inshore branch and 20–40 m below the surface at the offshore branch. Figure 6 shows the VMV positions in the subsurface layer; it also illustrates that the depth of the subsurface VMV in the inshore branch was deeper than that in the offshore branch. The difference can be explained by the combined effects of baroclinicity and wind friction. Assuming a relatively spatially homogeneous heat loss, different cooling occurs, due to the smaller heat capacity of the shallow coastal water compared to the deeper offshore waters; hence generating a northwestward horizontal density gradient leading to a northeastward thermal current (vertical current shear) according to the thermal wind relationship, resulting in an increasing of northeastward flow increasing upward. The northeasterly wind in winter weakens the northeastward TWC, particularly in the upper layer, which leads to the formation of the subsurface VMV. Therefore, the fact that the depth of the subsurface current core in the inshore branch is greater than that in the offshore branch indicates that a weaker baroclinicity or a stronger wind friction on the inshore branch than on the offshore branch."

**Line 304. ". . than on the . ."**

**Author's response:** Agree.

**Author's changes in manuscript:** Line 308: "than the offshore branch" was changed to "than on the offshore branch".

(3.2)

**Lines 334-335. ". . had a strong cross-shore component which means . ."**

**Author's response:** Agree.

**Author's changes in manuscript:** Line 339: "had a strong magnitude in the cross-shore direction" was changed to "had a strong cross-shore component".

**Lines 337-338. This implies that the ZMCC might be very wide, out to as much as 100m depth. This is a surprise to the reader because 30-100m depth was supposed to be the region for the TWC inshore branch.**

**Line 339. "episodically" means the TWC inshore branch is sometimes present and sometimes absent. For this, the fluctuations need to be more than "significant"; they sometimes need to be larger than the mean.**

**Author's response:** We agree that the statements of "ZMCC and TWC meet between the 30 and 100 m isobaths" and "episodic occurrence of TWC inshore branch" at the first sight are contradictory. However, one has to clearly distinguish between the mean flow and the variability here. Our statement that the dominant region of the TWC inshore branch is located between the 30 and 100 m isobaths, was based on the wintertime climatological density distribution, and additionally, by our simulations of the climatological mean currents (Figs. 5 and 6). However, when considering the variability of the current system, it can be observed that strong fluctuations occur, leading to the situation that the TWC becomes episodically and the ZMCC might be very wide for certain periods, as observed at the site off the Zhe-Min coast (Fig. 4). To clarify these points, we revised the following statement.

**Author's changes in manuscript:** Line 341-348: "In the inshore area, the fluctuation was located in a wide region between the 30 and 100 m isobaths, where the southwestward flowing ZMCC and the northeastward directed TWC meet. As deduced from the standard deviation, the currents fluctuated significantly in the alongshore direction, indicating that the TWC inshore branch occurred episodically."
**was changed to** "In the inshore area, the fluctuations were influencing a wide region between the 30 and 100 m isobaths, with a magnitude sometimes being larger than the mean flow (Fig. 5a). These strong fluctuations led to an episodic occurrence of the TWC inshore branch, as observed at the site off the Zhe-Min coast (Fig. 4, high temperature). When the TWC inshore branch was weakened due to these fluctuations, the ZMCC might even dominate a wide region outside of the 100 m isobath, especially at the surface (Fig. 4, low temperature.)".

**Lines 347-348. Better ". . Richardson et al., 2013), i.e. where the Richardson number equals the critical value 0.25 in this paper . ."**

**Author's response:** Agree.

**Author's changes in manuscript:** Line 357: Changed to ". . Richardson et al., 2013), i.e. where the Richardson number equals the critical value 0.25 in this paper . .".

**Line 353. Omit "Hence". [Correlation does not show what causes what. The rest**

**of the sentence is OK because wind and cooling are forcings.]**
**Author's response:** Thanks.
**Author's changes in manuscript:** Line 362: "Hence" was deleted.

**Line 359. "account for 54% . . (Fig. 9), associated . ." ["which" can only refer to the object immediately before].**
**Author's response:** Thanks.
**Author's changes in manuscript:** Line 368: Changed to "account for 54% . . (Fig. 9), associated . .".

**Line 365. Omit "great".**
**Author's response:** Agree.
**Author's changes in manuscript:** Line 374: Omitted "great".

**Lines 369-372. There is some repetition here.**
**Author's response:** We have now revised the statement.
**Author's changes in manuscript:** Line 378-382: "The spatial pattern of the second EOF mode (EOF2, Fig. 9b) shows a synoptic fluctuation in the inshore area. The fluctuation mainly varied in the alongshore direction, which indicates the episodic occurrence of the TWC inshore branch. The area with alongshore fluctuation (Fig. 9d) larger than 0.1 m/s was located between the 30 and 100 m isobaths, which demonstrates that the TWC could also episodically affect this area" **was changed to** "The spatial pattern of the second EOF mode (EOF2, Fig. 9b) shows a synoptic fluctuation in the inshore area. The area with alongshore fluctuation (Fig. 9d) larger than 0.1 m/s was located between the 30 and 100 m isobaths, which demonstrates that the TWC could episodically affect this area".

**Line 374. "great" -> "larger".**
**Author's response:** Agree.
**Author's changes in manuscript:** Line 383: "great" was changed to "larger".

**Line 390. "would then replace the . ."**
**Author's response:** Agree.
**Author's changes in manuscript:** Line 399: "which replaces" was changed to "which would then replace".

**Line 391. "Feb. 14-18"? This sentence is not very convincing. What about the deepening on 7-10 February?**
**Author's response:** Yes, you are right. Probably due to the fact that the wind shown in Fig. 11 is regionally averaged, while the mixed layer depth given in Fig. 8 represents conditions at a specified location, there is not such a clear connection between these two parameters. Therefore, we deleted this unconvincing statement.
**Author's changes in manuscript:** Line 399-401: deleted "Together with the effect of net surface heat flux, the stronger northerly monsoon during Jan. 5-13, Jan. 19-25 and

Feb. 16-18 causes the deepening of the mixed layer (P2, Fig. 8)."

(4 Discussion)
**Line 443. Omit first "The". "winters" – please state here if this is December – March or January – February.**
**Author's response:** We have now revised the statement.
**Author's changes in manuscript:** Line 452: "The simulated results in the winters" was changed to "Simulated results in the winters (December-March)".

**Line 446. ". . were present in all winters from 2009 to 2013."**
**Author's response:** Agree.
**Author's changes in manuscript:** Line 455-456: Changed to ". . were present in all winters from 2009 to 2013".

**Line 452. ". . manifested by two . ." [delete "with"]**
**Author's response:** Agree.
**Author's changes in manuscript:** Line 461: deleted "with".

(4.2)
**Line 538. Do you mean "exchange" or just that water between the 30 and 100m isobaths may be either ZMCC or TWC water.**
**Author's response:** Thanks. It should read: "water between the 30 and 100m isobaths may be either ZMCC or TWC water".
**Author's changes in manuscript:** Line 546-548: "water exchange between the ZMCC water and the TWC water exists in the area between the 30 and 100 m isobaths" was changed to "water between the 30 and 100 m isobaths may be either ZMCC or TWC water".

**Lines 543-544. ". . considered, not short-term . ." [Omit "in the relation . . and no"]**
**Author's response:** Agree.
**Author's changes in manuscript:** Line 552-553: "only wind-induced synoptic fluctuations are considered in the relations to the episodic events and no short-term extreme storm events" was changed to "only wind-induced synoptic fluctuations are considered, not short-term extreme storm events".

**Lines 558-559. "(Fig. 5a). Thus the offshore transports . ."**
**Author's response:** Agree.
**Author's changes in manuscript:** Line 569: "This also indicates that" was changed to "Thus".

(5 Conclusions)
**Line 584. ". . was nearly balanced . ."**

**Author's response:** Agree.

**Author's changes in manuscript:** Line 594: "was balanced" was changed to "was nearly balanced".

**Figures; generally it is better have units named on color scales.**
**Author's response:** Thanks. We have now added units on all the color scales.

**Author's changes in manuscript:** added color scales in Figure 1 (line 894), Figure 5 (line 922), Figure 8 (line 944) and Figure 12 (line 969).

**Figure 1 caption (lines 779, 875) "derived from the GDEM . ."**
**Author's response:** Agree

**Author's changes in manuscript:** Line 792, 895: "derived the GDEM" was changed to "derived from the GDEM".

**Figure 2 caption (lines 788, 884) "lines (right) show . ."**
**Author's response:** Agree.

**Author's changes in manuscript:** Line 801, 905: "lines show" was changed to "lines (right) show".

**Figure 5 caption (lines 807, 906) "ECS, shown by color. Sections .."**
**Author's response:** Agree.

**Author's changes in manuscript:** Line 820, 927: "in the ECS" was changed to "in the ECS, shown by color".

**Figure 7 caption. Explain black arrows. Lines 818, 919: ". . the branches' representative .."**
**Author's response:** Thanks. We have now added an annotation for the black arrows.

**Author's changes in manuscript:** Line 830, 939: "Current standard deviation" was changed to "Current standard deviation (black arrows)". Line 833, 942: "their representative points" was changed to "the branches' representative points".

**Figure 8 caption (lines 821, 923) ". . currents (m/s, shown by color scale) for . ."**
**Author's response:** Agree.

**Author's changes in manuscript:** Line 835, 945: Changed to "currents (m/s, shown by color scale)".

**Figure 9 caption (lines 827, 930) ". . (f) EOF2 cross-shore component (all shown by color scale). The 30," Explain black arrows.**
**Author's response:** We have revised the caption and added an annotation for the black arrows.

**Author's changes in manuscript:** Line 842-843, 953-954: "(f) EOF2 cross-shore component" was changed to "(f) EOF2 cross-shore component (all shown by black arrows with the color representing the magnitude)".

**Figure 10 caption (lines 831, 935) "along" -> "across" or "at" (twice).**
**Author's response:** Agree.
**Author's changes in manuscript:** Line 847, 853, 959, 966: "along" was changed to "across".

**Figure 12 caption (lines 841, 947) ". . Eq. (5) (shown by the color scale; units: . ."**
**Author's response:** Agree.
**Author's changes in manuscript:** Line 857-858, 971-972: "(units: $10^{-4}$ m$^2$/s$^2$)" was changed to "(shown by black arrows with the color representing the magnitude; units: $10^{-4}$ m$^2$/s$^2$)".

**Figure 14 caption (lines 850, 959). What do the black arrows represent on the EOF plots?**
**Author's response:** We have revised the caption and added an explanation for the black arrows.
**Author's changes in manuscript:** Line 867-871, 984-988: Changed to "The black arrows in the upper panels show the velocity (m/s) in the layer of VMV with the color representing the current speed. The two blue arrows with label IB and OB represent the flow axes of the inshore branch and offshore branch, respectively. The black arrows in the middle panels and bottom panels represent the EOF components (m/s) with their magnitude represented by color scales".

**Figure 15 caption (lines 858, 968). This is confusing. If P1 varies in time it could be anywhere and therefore useless.**
**Author's response:** We have now revised Figure 15 and its caption.
**Author's changes in manuscript:**
**Line 990:** added two lines (L1 and L2) in Figure 15.

[revised manuscript text omitted]

---

## Author Response (AR3)

| Ms. Ref. No.: | os-2016-70 |
|---|---|
| Title: | Synoptic fluctuation of the Taiwan Warm Current in winter on the East China Sea shelf |
| Journal: | Ocean Science |

Dear Dr. John M. Huthnance,

Thanks very much for your comments and assistances in editing our joint manuscript. We have made all the related corrections on the manuscript according to your comments.

Appended to this letter is the "Response to the referee" and a "marked-up manuscript". The "Response to the referee" contains our point-by-point responses to the comments raised by you. The "marked-up manuscript" tracks our changes.

Yours sincerely,

Jiliang Xuan, Daji Huang, Thomas Pohlmann, Jian Su, Bernhard Mayer, Ruibin Ding, Feng Zhou
January 13, 2017

**Response to the referee**

**Line 26. Better to omit "the" before "geostrophic".**
**Author's response:** Agree.
**Author's changes in manuscript:** Line 26: "the geostrophic balance" was changed to "geostrophic balance".

**Lines 89-90. "negative density anomalies" (line 89) does not seem to agree with "less low-density water was transported to . ." (line 90).**
**Author's response:** Thanks. Apparently it is "more low-density water" which was transported to the ECS shelf and causes the negative density anomalies.
**Author's changes in manuscript:** Line 90: "less low-density water" was changed to "more low-density water".

**Line 124. "wintertime TWC synoptic . ." (I think you intended this anyway).**
**Author's response:** Agree.

**Author's changes in manuscript:** Line 124: "wintertime synoptic fluctuations" was changed to "wintertime TWC synoptic fluctuations".

**Lines 296-297. Better ". . resulting in an upward-increasing northeastward flow. The northeasterly . ."**
**Author's response:** Agree.
**Author's changes in manuscript:** Line 296-297: "resulting in an increasing of northeastward flow increasing upward" was changed to "resulting in an upward-increasing northeastward flow".

**Line 299-300. Better ". . indicates weaker baroclinicity or stronger wind . ."**
**Author's response:** Agree.
**Author's changes in manuscript:** Line 300: "indicates that a weaker baroclinicity or a stronger wind friction" was changed to "indicates weaker baroclinicity or stronger wind friction".

**Line 333. Better ". . magnitude that was sometimes larger . ."**
**Author's response:** Agree.
**Author's changes in manuscript:** Line 333: "a magnitude sometimes being larger than" was changed to "a magnitude that was sometimes larger than".

**Line 336. "outside of the 100 m isobath". Do you mean "out to the 100 m isobath"?**
**Author's response:** Thanks.
**Author's changes in manuscript:** Line 336: "outside of the 100 m isobath" was changed to "out to the 100 m isobath".

**Line 337. I do not understand "low temperature". The figures do not show temperature. You could omit it?**
**Author's response:** We have now revised the statement.
**Author's changes in manuscript:** Line 335: "Fig.4, high temperature" was changed to "Fig.4, red color"; Line 337: "Fig.4, low temperature" was changed to "Fig.4, blue color".

**Figure 7 and caption (line 921). Current is a vector and its standard deviation has two components which can be represented by an ellipse but not fully by an arrow. Are the black arrows perhaps the ellipse (semi-) major axis?**
**Author's response:** Thanks. As you said, the black arrows indicate the major axis of the ellipse of the current standard deviation.
**Author's changes in manuscript:** Line 924-926 and 814-816: we revised the caption as following: "The black arrows indicate the major axis of the ellipse which represent the standard deviation of the current. The color shading shows the respective magnitude.

**Figure 13 and caption (lines 960, 961). The main text line 430 refers to "northwestward" Coriolis force and seems correct but this is not the "alongshore" direction stated in the figure caption.**

**Author's response:** Thanks. We have corrected the "alongshore" direction and the "cross-shore" direction in Figure 13 and changed the legends accordingly.

In addition, we added a pointer to indicate the alongshore and cross-shore directions in Figure 13 (marked in grey) in order to provide a better impression of dynamical effects in earth coordinate system, e.g., a negative value of the Coriolis force (Fig. 13c, black line) indicates a northwestward directed Coriolis force.

[Figure]

**Author's changes in manuscript:**
Line 963: we added grey pointers in Figure 13.
Line 964-967 and 847-850: we switched the statements of "alongshore" and "cross-shore". We also added a notation for the grey pointers: "
[revised manuscript text omitted]